# Potential Phase Change Materials in Building Wall Construction—A Review

**DOI:** 10.3390/ma14185328

**Published:** 2021-09-15

**Authors:** Abdulaziz Kurdi, Nasser Almoatham, Mark Mirza, Thomas Ballweg, Bandar Alkahlan

**Affiliations:** 1The National Centre for Building and Construction Technology, King Abdulaziz City for Science and Technology, P.O. Box 6086, Riyadh 11442, Saudi Arabia; akurdi@kacst.edu.sa (A.K.); nalmoatham@kacst.edu.sa (N.A.); 2Fraunhofer Institute for Silicate Research ISC, Neunerplatz 2, 97082 Würzburg, Germany; mark.mirza@isc.fraunhofer.de (M.M.); thomas.ballweg@isc.fraunhofer.de (T.B.)

**Keywords:** phase change materials (PCMs), paraffin, fatty acid, hydrate salts, butyl stearate, encapsulation

## Abstract

Phase change materials (PCMs) are an effective thermal mass and their integration into the structure of a building can reduce the ongoing costs of building operation, such as daily heating/cooling. PCMs as a thermal mass can absorb and retard heat loss to the building interior, maintaining comfort in the building. Although a large number of PCMs have been reported in the literature, only a handful of them, with their respective advantages and disadvantages, are suitable for building wall construction. Based on the information available in the literature, a critical evaluation of PCMs was performed in this paper, focusing on two aspects: (i) PCMs for building wall applications and (ii) the inclusion of PCMs in building wall applications. Four different PCMs, namely paraffin wax, fatty acids, hydrated salts, and butyl stearate, were identified as being the most suitable for building wall applications and these are explained in detail in terms of their physical and thermal properties. Although there are several PCM encapsulation techniques, the direct application of PCM in concrete admixtures is the most economical method to keep costs within manageable limits. However, care should be taken to ensure that PCM does not leak or drip from the building wall.

## 1. Introduction

The storage of thermal energy for later use is a growing trend and is known as thermal energy storage (TES) in the literature. In TES systems, energy can be stored in a medium in the form of either “heat” or “cold” and is thus available for later use. It bridges the gap between the supply and demand of energy, mainly electricity. Ever-increasing energy consumption in various industrial sectors is causing global warming, and researchers are increasingly seeking newer ways to convert energy with a limited impact in terms of greenhouse gas (GHG) production [1]. According to the California Energy Commission, the potential deployment of TES systems could result in a reduction in NO_x_ levels of 560 tons and a reduction in CO_2_ emissions of 260,000 tons from the building sector across the state [2]. If we consider all the associated costs in the building industry, such as the ongoing costs of heating and cooling, the costs of fossil fuel electricity generation, and the associated greenhouse gas emissions (CO_2_, SO_2_, NO_x_, CFCs), TES, along with other conventional and unconventional energy storage systems, is very promising for the sustainable development of the building industry [3]. Research on the efficient use of solar thermal energy has been established worldwide, because it is one of the cleanest energy sources and is abundant in nature. However, limiting factors include its absence at night and on cloudy days, posing a major challenge for a continuous energy supply. One answer to this challenge is to develop a medium that is sensitive enough to store thermal energy in the absence of sunlight. In this regard, phase change materials (PCMs) have found new applications in the construction industry, especially in green buildings. Green buildings are referred to as energy-efficient buildings, using much less traditional energy (e.g., electricity) while maintaining the same level of comfort. In short, PCMs tend to be characterized by a phase transition that occurs at a more or less constant temperature. Here, the phase transition is not limited to melting, but can also be evaporative, for example. It is interesting to note that PCMs are neither new nor exotic materials, as is often claimed in the literature [4]. PCMs have low density and form a viscous/semi-viscous mass when melted (compared to free-flowing water), thus avoiding the problem of the use of heavy materials in building design and construction.

Several studies on the development of PCMs, as well as their incorporation into the building envelope, have been reported in the literature. Nevertheless, it is difficult to obtain a good understanding because most of the reported work is not scientifically linked to previous work and is disorganized. There are also a number of reviews in the literature, but they are disjointed. The main reason for this is that researchers have tried to combine all the available information into one paper, ignoring the complexity and vastness of the topic. Therefore, it is necessary to address each aspect of PCMs separately. With respect to phase change materials, two different aspects are considered in this review. They are:Phase change materials themselves;How to use them in building construction.

When incorporating PCMs into building envelopes, each building envelope requires PCMs of a specific nature, shape and size. Considering this, the aim of this article is to provide a comprehensive overview of PCMs applied in the building envelope. Therefore, they will be referred to as “smart walls” in the following. In this study, all the possible PCMs available for this purpose have been critically analyzed, and the related information, such as the development of PCMs, their physical and thermal properties, and their integration into the building envelope, has been presented. In this way, industry partners and researchers can benefit from this comprehensive overview and overcome the associated limitations and drawbacks to meet the future challenge of sustainable development. In addition, the associated challenges in the use of PCMs, in terms of materials and methodology, are also discussed. The new feature of this study is to assess the advantageous usage of PCMs in the Middle East region in general and Saudi Arabia in particular, in terms of lower energy consumption, energy efficiency, and low-cost home insulation systems. The current study will enable engineers to select the particular PCMs for a given application, as well as helping researchers to carry out more innovative research work in this area.

## 2. Impact of PCMs on Building Construction

As mentioned earlier, latent heat storage provides a higher density of energy storage than sensible heat storage. In addition, the temperature variation is small for the former. The scenario can be modeled on that of a thermal switch. Once the switch is turned on, i.e., the PCMs reach their melting point, the temperature is kept constant until the material is completely melted. This melting process allows a large amount of heat to be absorbed while keeping the system temperature constant at the melting temperature of the respective PCMs [5]. Since PCMs store heat or cold for later use, they can thus regulate or dampen temperature fluctuations within a building, automatically leading to energy savings. In other words, PCMs provide a virtual building mass. If the room temperature exceeds the melting temperature of the PCM, it melts and absorbs heat. Later, when the outside temperature rises, the interior of the room will not easily reach a high temperature. In colder weather, the PCMs release the heat. A brief justification for the potential use of PCMs is shown in Figure 1, in which the thickness of the thermal mass is compared to traditional building materials such as gypsum, wood, concrete, sandstone and brick. Thus, the use of PCMs not only reduces the thermal mass, but also reduces the footprint of the building and provides significantly more usable space inside the buildings.

As recently reported by Lagou et al. [4] in their numerical and simulation work, in addition to the PCM itself, the optimal positioning of the PCM in a given building envelope also plays a significant role. According to the authors [4,5,6], the optimal location to incorporate PCMs is the interior edge of the building and also depends on the geographical location. In addition, the energy payback time for PCM incorporated building elements is less than 7 days/m^2^. Such promising data on energy payback time creates a motivation to further incorporate PCMs into building wall construction. The following section will present the classification and general properties of PCMs for building wall applications.

## 3. Classification and Properties of PCMs and Their Application in Building Walls

PCM is not a single material, but a group of materials that have a number of specific properties. Broadly speaking, PCMs are divided into the following categories:Organic;Inorganic; andEutectic, as explained in subsequent sections.

### 3.1. Organic PCMs

Several factors should be considered when developing PCMs. However, the most important of these are their cost, thermal conductivity, latent heat content and freezing/melting range. Examples of potential organic PCMs are waxes, oils, fatty acids and polyglycols. These types of organic PCMs are in the form of a long molecular chain with a carbon backbone. The melting point of such a material is determined by the length of the carbon molecular backbone. Generally, the longer the chain, the higher the melting point. Since these materials originated as single-chain molecules, this makes their melting point more specific, along with an increase in their latent heat content. Examples include pure linear hydrocarbons, which require complex processing and are therefore expensive. On the other hand, naturally occurring fats such as vegetable oils and animal fats are economically attractive compared to synthetic fats. However, these naturally occurring fats consist of a wide range of molecules with different chain lengths. Consequently, instead of a sharp melting/freezing point, these materials exhibit a temperature range where complete melting/freezing occurs. Since PCMs must operate in a narrow temperature band for efficient TES operation, these naturally occurring fatty materials are not suitable for this purpose. In addition, due to their highly oxidizing behavior over time, these types of fats are fire hazards and can only be used with the use of flame-retardants.

On the positive side, organic PCMs do not suffer from subcooling problems and thus avoid phase separation over time. In addition, they are chemically stable, non-corrosive and non-toxic, but have the disadvantage of relatively low thermal conductivity and potential flammability. Organic PCMs can be further classified as 

Paraffin andNon-paraffin types.

### 3.2. Inorganic PCMs

Compared to organic PCMs, inorganic PCMs are non-reactive (fire-resistant) and have higher latent heat content and high thermal conductivity. Their disadvantages are their corrosive nature and the fact that they suffer from supercooling and phase separation over time [7]. As reported by Mehling and Cabeza [8], additional measures are required to counteract these drawbacks, through the introduction of additives, additional nucleating agents, the dispersion of highly thermally conductive material such as fused perlite or metal dusts, and a general microencapsulation technique [9]. Inorganic PCMs are mostly water-based hydrated salts with a freezing/melting temperature above 0 °C [10]. It is common practice to use a mixture of salts to fine-tune the phase change temperature, i.e., the freezing/melting temperature. It is also possible to use the same salt in different concentrations (in water). Usually, the water chemically combines with the salts to form a crystalline structure commonly known as a “hydrated salt”. Interestingly, some hydrated salts can contain up to 50% water, although their physical appearance is in solid crystal form and usually has a distinct color, depending on their water content. When hydrated salts are heated, the water portion of them is separated and the salt dissolves in this water. During this process, the system absorbs the heat flux in the form of latent heat. Exactly the opposite takes place during freezing, as heat escapes from the system. The practical application of this theory produces PCMs that can melt/freeze at up to 117 °C. In other words, these PCMs are frozen at temperatures higher than the boiling point of water. Congruent hydrated salts are transparent when melted and the chemical composition of the melt phase is the same as that of the solid phase before melting. Others are semi-congruent, i.e., they form a different hydrated salt with a lower melting point upon melting. Recently, Guo et al. [5] has proposed inorganic VO_2_ as a “smart” phase change material for building applications. This material changes its phase upon external thermal excitation at around 67 °C, which is useful in external walls. This material, used as a coating, is highly manufacturable and scalable with high-throughput.

### 3.3. Eutectic PCMs

Eutectic PCMs are a mixture of the above-mentioned organic and/or inorganic PCMs and can be sub-divided into different groups. The groups include:Organic–organic,Inorganic–inorganic andInorganic–organic.

Eutectic PCMs are congruent in nature and occur in crystal forms [11]. In its simplest form, the theory behind eutectic PCMs is as follows. When salts of any kind are added to an aqueous medium, the freezing point of the water is lowered. This is the reason why gritting materials are spread on icy/snowy roads in the winter to keep the roads open. As salts are added to the solution, the freezing temperature drops further. At a certain composition, the mixture takes the form of a slush. However, under certain circumstances, such as a mixture of certain salts at a certain concentration, it melts and freezes translucent at a certain temperature. During the melting/freezing process, the system releases or stores heat in the form of latent heat. The composition of the mixture at this point is called the eutectic composition and this particular temperature is called the eutectic temperature. A brief comparison of PCMs in terms of their classification and common properties is given in Table 1.

In this regard, the most studied PCMs are hydrated salts, paraffin/non-paraffin waxes, animal/plant-based fatty acids, and eutectic PCMs combining any of the above materials. The details and properties of these materials are described below.

#### 3.3.1. Paraffin Waxes

Paraffin wax (PAR) is a by-product of petroleum refining and can be used as a value-added material as a PCM. It is a mixture of several linear alkyl hydrocarbons. The melting point of PAR is comparable to that of salt hydrate, with reasonable latent heat and without the problem of supercooling associated with salt hydrate. A major disadvantage of PAR is its high flammability; therefore, its use is only recommended in combination with fire-retardant fillers. The chemical formula of paraffin wax is C_n_H_2n+2_, that is, an aliphatic chain (e.g., C_3_H_8_, C_5_H_12_, C_7_H_16_) as shown in Figure 2 [24]. Commercial paraffin waxes are mixtures of different waxes with a wide range of melting temperatures and are usually cheap, with acceptable heat storage densities in the range of ~200 kJ/kg or 150 MJ/m^3^. However, the main disadvantage is that they have a low thermal conductivity coefficient (~0.2 W/m °C). To overcome this, the addition of fillers is necessary.

The fillers not only increase the thermal conductivity but can also retain a larger volume of PCMs therein. Common fillers include metal dust/particles, molten dolomite/perlite or metal inserts such as finned tubes and aluminum chips [13]. Commercial paraffin, as opposed to pure paraffin, is most commonly used in PCMs due to its high cost, with a melting temperature of about 55 °C [25,26]. Farid et al. [27] studied commercial paraffins with melting points of 44, 53 and 64 °C, which have latent heat densities of 167, 200 and 210 kJ/kg, respectively. The first result is promising in terms of their ability to maintain the temperature level in the comfort zone. Detailed technical information on this can be found in the literature by Himran et al. [21], Faith [28] and Hasnain [13]. A list of the thermo-physical properties of the most common paraffin waxes used in the construction of building walls is given in Table 2.

#### 3.3.2. Fatty Acids

Fatty acids are attractive candidates for TES systems such as PCMs for latent heat energy storage in space heating applications. As reported by Feldman et al. [39], the physical and thermal properties of fatty acids (capric, lauric, palmitic and stearic acids) and their binary mixtures meet the requirements for use as PCMs. The melting point of the fatty acid group PCMs ranges from 30 °C to 65 °C and the latent heat content varies from 153 to 182 kJ/kg. In a parametric study of palmitic acid PCMs by Hasan et al. [40], the behavior of such PCMs including the phase transition, solid/liquid interphase in the mixture, freezing/melting temperature and the heat flux rate in a TES system consisting of circular tubes was described in detail. Among the various fatty acids, capric and lauric fatty acids are mainly used in low-temperature TES systems, as reported by Dimaano et al. [41], with a freezing/melting point of about 14 °C. Depending on the composition, the latent heat content is about 113–133 kJ/kg. As mentioned earlier, although a number of materials in this category have been studied at the laboratory scale, only a handful of them have been explored to their full potential, so there are vast opportunities for further research in this area. Examples of such laboratory PCMs include dimethyl sulfoxide, with a melting temperature of 16.5 °C and a latent heat content of 86 kJ/kg [16]; maleic anhydride, with a melting temperature of 52 °C and a latent heat content of 145 kJ/kg [42], etc. Like PAR, fatty acid also has a long chain of molecules, as shown in Figure 3. Depending on the location of the double bonds in fatty acids, they can be further classified as:Saturated andUnsaturated fatty acids.

A list of common fatty acid PCMs with their respective thermo-physical properties is given in Table 3.

#### 3.3.3. Hydrated Salts

Hydrated salts are attractive materials for use as PCMs because they have high thermal energy storage density (~350 MJ/m^3^), along with a high thermal conductivity coefficient (~0.5 W/m °C) and a modest cost. Hydrated salts are mainly crystalline salts that contain a certain amount of water in crystalline form and therefore have different colors depending on the water content. For example, Glauber’s salt (Na_2_SO_4_.H2O) contains 44% Na_2_SO_4_, whereas the water content is higher than the salt content at 56%, although they appear in a solid form at room temperature and pressure. Glauber’s salt was one of the first studied PCMs, reported in the 1950s [12,17], with a melting temperature of about 32.4 °C and a latent heat content of 254 kJ/kg (377 MJ/m^3^). The physical states of some common hydrated salts at room temperature are discussed below. Despite their high thermal conductivity, problems related to supercooling and phase separation are a major challenge for their wide application. To deal with this problem, the “extra water principle” is the most commonly used solution to prevent the formation of heavy anhydrous salts, as mentioned by Biswas et al. [63]. However, the use of extra water extends the melting/freezing temperature range, which negatively affects the thermal energy storage density of the material. Some researchers have proposed the use of bentonite clay as a thickening agent, but this introduces another challenge, namely, the reduction of the crystallization rate, as well as the heat transfer rate, due to the low thermal conductivity coefficient of the system. Borax can also be used as a nucleating agent to prevent supercooling [12]. In general, hydrated salts suffer from the problems of supercooling, nucleation and phase segregation and therefore require the use of encapsulation, along with thickening and nucleating agents. A list of hydrated salts with their respective thermos-physical properties suitable for wall construction can be found in Table 4.

#### 3.3.4. Butyl Stearate (BS)

Butyl stearate (BS) is another organic PCM that has been described in the literature as one of the best-known PCMs. It has a melting point of 17 °C–22 °C [80] and the linear chemical formula is CH_3_(CH_2_)16COO(CH_2_)3CH_3_. BS can be used both through the microencapsulation technique and directly as an admixture in concrete. Liang et al. [81] reported the microencapsulation of butyl stearate in poly-urea microcapsules using the interfacial poly-condensation method with a particle diameter of about 20–35 µm with toluene 2,4-diisocyanate (TDI) and ethylene diamine (EDA) as backbone monomers. This microencapsulated butyl stearate shows a melting temperature of about 29 °C with a latent heat of fusion of about 80 J/g [81]. Examples of such microencapsulated BS in the form of optical images are shown in Figure 4. 

Microencapsulated butyl stearate can be used as an additive for interior or exterior coatings or can be included in insulating materials such as foams. Melamine formaldehyde resin is widely used in industry. It has high mechanical strength and desirable thermal resistance properties. The thermal decomposition temperature of melamine formaldehyde is above 300 °C, which is suitable for use as a wall-building material [82]. A number of different polymers and monomers can be used to encapsulate PCMs, namely, amine resins, poly-urea, polyurethane, polyester, polyamide, etc. These polymers form a shell-like container, which then contains PCMs inside it, and the process is referred to as poly-condensation in the literature [81]. For example, Liang et al. [81] used poly-urea as the shell wall material. The particle diameter of the microcapsules in their study was about 20–35 μm [82]. In another work by Hua et al. [83], butyl stearate was microencapsulated over poly-urea polymer. They prepared microcapsules with a size of 4.5–10.2 μm [82]. Bendic and Amza fabricated poly-methyl meta-acrylate (PMMA) microcapsules with butyl stearate as the core with an average diameter of 95 μm [21]. Oliver et al. [84] reported on the role of pH in the composition and thermal stability of melamine microcapsules with butyl stearate on their fabricated microcapsules with a diameter of 20–50 μm [22]. On the other hand, Cellat et al. [80] reported the direct incorporation of BS as an admixture in concrete to improve the thermal performance of building walls. A list of butyl stearate PCMs with their respective thermo-physical properties suitable for wall construction is given in Table 5.

As can be seen from above-mentioned data and properties of PCMs, there are a number of PCMs that can be a candidate for building wall applications. However, each PCM is unique in nature and comes with respective benefits and challenges, as described in the next section.

## 4. Challenges Associated with the Use of PCMs

For the successful application of PCMs, some of their inherent limitations should be considered and properly addressed. The main limitations arise in terms of the thermal conductivity coefficient, changes in heat storage density and the maintenance of the initial system’s efficiency over time, as well as phase segregation and subcooling, as described below.

### 4.1. Phase Segregation and Supercooling

As mentioned earlier, when hydrated salt PCMs melt, supercooling and phase segregation occur before the next freezing cycle begins. Supercooling in PCMs is defined as a metastable state in which the PCMs remain in a liquid state (phase) even though the temperature is below their respective melting point. This supercooling and phase segregation suppress the efficiency of the PCMs over time and, in the worst case, can prevent the PCMs from solidifying at all. The initial heat storage density deteriorates over time and thus the efficiency decreases at an alarming rate. This is because hydrated salts are a congruent melting material, so melting is accompanied by reduced hydrate formation. This process is irreversible and since lower salt hydrates have a lower thermal energy storage capacity, the efficiency of the system decreases. Subcooling is also blamed for the decrease in efficiency of the hydrated salts. One of the solutions to this is the use of direct contact with the conducting environment, such as an immiscible heat transfer fluid [14,16,18,91,92]. The presence of additional heat transfer fluid causes agitation to occur in the system, which not only minimizes subcooling but also prevents possible phase segregation. Some researchers have used the ‘extra water principle,’ as described briefly earlier (Section 3.3), to avoid segregation and subsequent plugging. When developing additives/stabilizers to stimulate nucleation, the physical and chemical properties of the salts in question should be considered [19]. Ryu et al. [18] reported extensive research on the development and use of suitable stabilizing/nucleating agents that can be used in a range of PCM systems.

### 4.2. Stability of PCMs over Time

One of the hurdles to the widespread use of PCMs is their ability to maintain their physical, thermal and chemical properties over the time of use, i.e., over the thermal cycles to which they are exposed. Here, the effect of corrosion by PCMs in the system is also relevant, especially when they are macro encapsulated in a container [10]. PCM containers, both microencapsulation and macro encapsulation, must have sufficient physical and thermal stability as PCMs are subjected to repeated heating/cooling cycles. Kimura et al. [93] reported the use of NaCl in CaCl_2_.6H_2_O with additional water content, which can withstand up to 1000 heating/cooling cycles. On the other hand, as reported by Gibbs et al. [20], paraffin-based PCMs usually do not suffer because both thermal cycling and the use of the container do not affect their thermal properties. Unfortunately, no information on the corrosion of paraffin is available in the literature, and there are few reports on the corrosion of PCMs, and these lack details about this subject [8,94]. Despite all the above facts, hydrated salts are denser than the paraffin and thus the effective heat capacity per unit volume is high. As a rule of thumb, a 10% volume change in thermal cycling can be considered a minor problem [22]. Hawladar et al. [95] reported on the thermal stability of microencapsulated paraffin and confirmed its stability up to 1000 cycles, as shown in Figure 5.

It is important to note that there are few studies in the literature that address the thermal stability of PCMs over multiple cycles, and most of these papers are related to accelerated laboratory-scale thermal tests. Accelerated laboratory-scale tests do not always necessarily reflect the real-world scenarios of practical applications. A lack of detailed results on the thermal stability behavior of various PCMs is clearly visible in the literature, and thus further attention by researchers is required.

Cellat et al. [80] reported the thermal stability of microencapsulated BS PCMs subjected to a total of 1000 melting/freezing cycles via differential scanning calorimetry (DSC), as shown in Figure 6 [80]. After 800 cycles, the melting temperature and latent heat storage capacity changed from 21.0 °C to 20.8 °C and from 135.5 J/g to 105.1 J/g, respectively [80]. Although there was no significant change in melting temperature, the change in latent heat (22%) was significant, leading to the need for further investigation, as the current literature is not available.

### 4.3. Thermal Conductivity

In general, most PCMs suffer from a low thermal conductivity coefficient. To improve this, measures to increase their heat transfer capability are required to increase their thermal conductivity [96,97]. To increase the surface area of PCMs, which in turn increases the thermal conductivity of the TES system, a common practice is to impregnate a porous matrix with PCMs and then form a composite. These porous materials can be diatomaceous earth, silica, perlite, etc., as reported in the literature [98,99,100]. An example is shown in Figure 7, which demonstrates the inclusion of PCMs in the diatomaceous earth structure. The diatomaceous earth obtained (Figure 7a) contains impurities and therefore needs to be calcined in order to get rid of these impurities and open the pores of the structure (Figure 7b). After calcination, the organic impurities in the diatomaceous earth have been volatilized and thus the specific surface area has increased. After mixing with paraffin PCMs, the diatomaceous earth absorbs the paraffin in its pores, becomes coated with paraffin (Figure 7c,d), and looks like spheres from the outside.

Thus, the composite structure is similar to the core-shell structure, where the core is diatomaceous earth, and the shell is paraffin. The size of the composite spheres varies (5–20 µm) depending on the size of the diatomaceous earth, soaking time, temperature, the density of the PCM used and other related process parameters. In general, the composite exhibits a uniform structure due to the homogeneous absorption of kerosene into the diatomaceous earth and has sufficient mechanical strength for handling and standing in the application [100]. So far, different kinds of PCMs and their associated properties has been discussed. However, identifying an optimum PCM does not necessarily mean that their application in building walls will offer the best performance, as the integration of such PCMs is a challenge itself. The next two sections (Section 5 and Section 6) summarize various ways of integrating PCMs into building wall applications.

## 5. PCM Integration into the Building Envelop

Although PCMs can be integrated into various building envelopes, as reported in the literature [19,101], in this paper we will focus mainly on the integration of PCMs into building walls. In buildings with multiple floors, the roof space becomes tight due to the space required for solar panels and air conditioning units. Therefore, to make the best use of the available space, building walls are the second-best place to install PCMs and are referred to here as “smart walls”. During the phase transition, PCMs melt into viscous/semi-viscous forms. Therefore, to avoid leakage, they must be properly enclosed in protective containers in a process called encapsulation. To achieve the best possible performance, PCMs must be bonded to the inner surface of the walls whenever possible. This thermally couples the PCMs to prevent the loss of thermal conductivity. Thus, the traditional technique of maintaining an air gap between the inner and outer layers of the walls is not required. Lane et al. [102,103] identified over 200 potential PCMs, covering a wide operating temperature range (10 °C to 90 °C) which can be used for encapsulation. Two types of encapsulation are widely used in commercial applications, namely:Micro-encapsulation andMacro-encapsulation, as discussed below in detail.

### 5.1. Micro-Encapsulation

Microencapsulation of PCMs refers to the incorporation of PCMs in microscopic shells, where the shell consists of polymers/monomers and one or more PCMs in colloidal form constitutes the core substance [81]. Depending on the application, the shell can be polymeric or inorganic in nature. The microencapsulation of hydrated salt PCMs (e.g., CaCl_2_.6H_2_O) in a polyester-resin micro-container has been promising, and the application of these PCMs to building walls has been reported in the literature [104]. The final product can take the form of extruded films, thanks to the solvent exchange technique. This process can achieve microencapsulated PCMs with about 40% PCM retention. These microencapsulated PCMs films show good mechanical and thermal constancy under cyclic freeze-melting conditions. As reported by Royon et al. [104], PCMs with water phase (i.e., hydrated salts) can be contained in polyacrylamide polymers. This polymer forms a three-dimensional network of macroscopic polymer chains and acts like a net to trap PCMs within it. The gap between the chains is small enough to prevent the leakage of PCMs due to absorption, as shown schematically in Figure 8.

Paraffin wax can also be encapsulated by forming an inorganic shell around it, e.g., a shell of calcium carbonate (CaCO3), using the self-assembly method shown schematically in Figure 9. First, oily paraffin is diffused in an aqueous solution, together with nonionic surfactants, to form a stable emulsion. Then, the chains of surfactants, which are hydrophobic by nature, are oriented to the oil droplets. At the same time, the hydroxyl groups of the surfactants, which are hydrophilic by nature, combine with the water molecules and stay away from the oil droplets. The layer of surfactants covers the surfaces of the oil droplets, forming kerosene micelles. Subsequently, Ca^2+^ ions are bound by the hydroxyl groups of the surfactants when droplets of CaCl_2_ are added to the same solution. This process is called the complexation process. Finally, CaCO_3_ is formed because of a precipitation reaction by introducing Na_2_CO_3_ into the same emulsion system. This CaCO_3_ is not soluble in the system and acts as a shell encapsulating the oily phase (PCMs) of the emulsion. Since this CaCO_3_ forms directly on the surface of the paraffin micelles, the process is referred to in the literature as the self-assembly process.

Details of the structure and appearance of microencapsulated paraffin in the initial state are shown in Figure 10 [105]. As can be seen from Figure 10, the morphology of the encapsulated PCMs varies from spherical with some coagulation to shell-like shapes and the diameter is about 1–5 µm. Besides the different paraffin loading, the processing temperature, the presence of additives and the presence of surfactants play a crucial role. As can be seen in the TEM images (Figure 10), a representative core-shell formation of the microcapsules with a shell thickness of about 0.14–0.5 µm is confirmed [105].

As with the encapsulation of paraffin in a CaCO_3_ shell using the self-assembly process mentioned above, paraffin or other PCMs can also be encapsulated by a polymer shell. As an example, the process of microencapsulation of n-octadecane PCMs with poly-urea shells is shown in Figure 11, and the corresponding SEM images of the microencapsulated PCMs are presented in Figure 12.

All this evidence confirms the possibility of various microencapsulation processes, which provide easy and versatile methods for the use of PCMs. The advantages of the microencapsulation of PCMs in various media include increased surface area, resulting in improved heat transfer surface, prevention of the contact of PCMs with the environment, which helps prevent fouling and oxidation of PCMs, and overall maintenance of the volume of the storage materials, which ensures the maximum achievable efficiency of the system over the lifetime of the component [81]. Different kinds of novel and efficient micro-encapsulation techniques are still under investigation and are being reported by researchers. For example, Ballweg et al. [80] has reported ultra-violet (UV)-based micro-encapsulation techniques for hydrated salts and paraffin waxes, which represent a relatively quick and efficient process. However, further investigation on the long-term stability of such micro-encapsulated PCMs are yet to be conducted.

### 5.2. Macro-Encapsulation

In contrast to microencapsulation, in macro-encapsulation, the PCMs are encapsulated in lifed-size containers that come in different shapes and materials, as shown in Figure 13.

The most widely used commercial microencapsulation techniques include Ø 2–3 mm spherical capsules, flat plates, metallic (stainless steel) spherical capsules and cylindrical bars filled with PCMs (Figure 13d). PCMs can also be encapsulated in bags made of conductive metallic materials such as thick aluminum foils/plates, as shown in Figure 14.

A list of PCMs that are suitable for micro-/macro-encapsulation, with their respective thermo-physical properties appropriate for wall construction, is given in Table 6.

## 6. PCMs for ‘Smart Wall’ Applications

As mentioned above, PCMs generally have a low coefficient of thermal conductivity and therefore require the use of an additional heat transfer medium, such as metal inserts, etc. This is a major disadvantage in their application and metal inserts are expensive. These disadvantages can be eliminated through the direct application of PCMs in wall surfaces. Thus, the use of wall panels containing PCMs in the building wall can provide smooth temperature variations. The large surface area of the wall allows for a higher thermal conductivity rate with the wall/room [22]. Another popular application is the use of wall panels that contain PCMs within themselves. Wallboards are widely available and economical, and the integration of PCMs into them is very promising. Kedl et al. [134] and Salyer et al. [135] presented the concept of a wallboard impregnated with paraffin wax by simply dipping the wallboard in paraffin wax. This process of PCM integration into the wallboard by simply dipping it in PCMs involves minimal cost, and can be scaled up to any size depending on the size of the wallboard [136]. Neeper et al. [137,138] studied the dynamics of the thermal behavior of gypsum wallboard impregnated with fatty acids and paraffin PCMs. According to these authors, the thermal storage capacity of the PCM-filled wallboards is sufficient to retain the thermal energy for the TES system. Stetiu et al. [139] used a computer-aided simulation technique to investigate the thermal performance of PCM-impregnated wall panels using finite element analysis (FEA) and reported a number of parameters that should be considered, such as the wetting of the PCMs on the wallboards, etc. Zhang et al. [106] investigated fatty acid impregnation and reported direct energy savings of 5–20% [53]. Athienitis et al. [75] reported the results obtained from both experimental and simulation approaches for a life-sized structure made of PCM-impregnated wallboards. The tests were conducted outdoors with rooms made of PCM gypsum boards containing about 25-wt% butyl stearate (BS) as the inner lining of the wall. According to their simulation work, which was supported by experimental results, the use of PCM in gypsum wallboard can reduce the room temperature by 4 °C during the day compared to pure gypsum board. For the direct applications of PCMs in building walls mentioned above, PCMs can be compressed into a sheet form and the surface wrapped and sealed with foil material to prevent leakage. One such commercially researched application is shown in Figure 15, developed by Energain^®^, which has been achieved by the Dupont de Nemours Society, UK [140].

Microencapsulated PCMs (e.g., paraffin) can be mixed directly with gypsum and used in the form of a panel, as shown in Figure 16 from Micronal^®^ produced by the BASF (Ludwigshafen, Germany). In this way, the wallboard can be used as a normal wallboard, which is much more energy-efficient than one without PCMs in it [140].

Since bricks are one of the most commonly used construction materials to form building walls, incorporating PCMs into bricks is an attractive way to use them with builders and end users. PCMs can be mixed with cement paste or used as a stand-alone PCM paste to fill the cavities of hollow bricks/blocks, as well as the interior plaster of the wall. Figure 17 shows different types of hollow bricks and the construction of a building wall with such bricks, together with the use of PCM paste to bond them. As recently reported by Gao et al. [110], PCM-filled hollow bricks improve the thermal behavior of building walls significantly. According to the authors [110], PCM-filled hollow bricks can reduce the attenuation rate from 13.07% to 0.92–1.93%, together with increasing the delay time from 3.83 h to 8.83–9.83 h. In other words, hollow bricks filled with PCM can reduce the peak heat flux from 45.26 W/m^2^ to 19.19–21.4 W/m^2^. In addition, inner cavities were the better choice for PCM and there was an extra phase-change extent of close to 90% in favor of the different outdoor thermal environment.

An example of the use of microencapsulated PCMs in the interior of plasterboard is shown in Figure 18.

A comparison of common wall-building materials in terms of their respective heat storage capacities is shown in Figure 19. The heat storage capacity of PCMs far exceeds that of all other building materials and is therefore promising as an efficient building material in the present and future.

### 6.1. Case Studies

According to Isaac et al. [142], based on their calculations using computer simulations, GHG emissions for the residential sector only will increase from 0.8 Gt C in 2000 to about 1 Gt C in 2020 and then more than double (2.2 Gt) by 2100 [103]. Much of this can be limited through the use of PCMs, which reduce the reliance on building heating/cooling, which in turn reduces fossil fuel-related electricity generation. Looking at the consumption of electrical energy, there is a staggering from 27 EJ (exajoule) of electricity consumption in 2000 to about 80 EJ (exajoule) in 2100 [103]. According to the International Institute of Refrigeration (IIR), electricity consumption by refrigeration and air conditioning systems in buildings accounts for about 15% of total electricity consumption [142,143,144]. Due to global warming and climate change, as well as increasing demands in developing countries, more and more buildings are being equipped with conventional air conditioning systems to provide the necessary comfort. In this sense, the widespread application of PCMs in the building sector could lead to savings, even if it can only cover a certain percentage of energy consumption. According to Jeon et al. [6], this could account for up to 55% of the total electricity consumption of buildings in cold climates such as in Korea [104], and the use of PCMs has great potential to reduce dependence on the electricity grid. As reported by Li et al. [14], heating, ventilation and air conditioning (HVAC) systems account for about 48% of building-related energy consumption in the United States [15], and a large portion of this can be managed by using PCMs in building wall construction. A summary of various experimental studies on the use of PCMs as wall construction materials in the last five years, as reported in the literature, is shown in Table 7, Table 8, Table 9 and Table 10.

### 6.2. Cost-Benefit Analysis of the Use of PCM

The design of buildings, whether commercial or residential, depends on a number of factors, and in the end, it is the consumer’s decision whether to choose PCMs over conventional insulation or a combination of both together with an efficient HVAC system. This decision also depends on local conditions and specific requirements. In Australia, for example, the cost of integrating PCMs is about AUD 55–110 per square meter, compared with about AUD 10/square meter of conventional insulation. If the building size is large enough, the cost of installing PCMs can break the budget. A number of commercial suppliers (e.g., BioPCM) offer readymade PCM products, in the form of sealed pouches/pallets. The weight of these pouches/pallets varies from 2.7 kg/square meter with a transition temperature in the range of 18 °C to 28 °C from solid to molten state. Typical recommended consumption by suppliers is 1–2 kg of PCM per cubic meter of room volume with a heat storage capacity in the range of 1.1 MJ/m^2^ (0.3 kWh/m^2^) [62]. The other potential benefits of using PCM as building materials are as follows:The use of PCMs may be advantageous in terms of comparable pricing to very expensive double/triple glazing.The use of PCMs has helped improve the energy rating of buildings from 7.8 to 9 stars through the use of PCMs together with conventional insulation [62].Since PCMs operate as passive heating/cooling materials, no power consumption is required compared to active heating/cooling (e.g., air conditioning). Thus, in the event of a power outage (both natural and system-related), the comfort level in the rooms can be maintained within the desired temperature range, at least for a certain number of hours. This makes it possible to keep the demand for electrical energy low while the repair of the system is carried out.

The use of PCMs also helps to reclaim more floor space, as the footprint is smaller (in the range of 50 mm) than conventional masonry, which can be up to 150 mm high [13]. PCMs are one of the available options, along with conventional insulation, efficient glazing, etc. [62]. A list of commercially available PCMs from different manufacturers with their respective thermal properties is given in Table 11.

## 7. Conclusions and Future Perspectives

In TES systems, PCMs play an important role and research on them is in the development stage, attracting interest and funding worldwide. The use of PCMs as building materials is a growing trend aiming to limit the energy consumption of buildings, and the direct use of PCMs in building walls is the most favorable. The most common PCMs fall into the category of organic, inorganic or eutectic PCMs with their respective advantages and disadvantages. Inorganic PCMs must be coupled with suitable binders (nucleation and thickening) to avoid phase segregation and sedimentation due to subcooling. For this purpose, considerable efforts are being made both in laboratories and in commercial companies to find versatile and efficient binders. Direct applications of PCMs are more desirable than that of micro/macro encapsulation, not only from an economic point of view, but also in terms of performance. There is still a long way to go to discover new types of PCMs and make them compatible with other common building materials in terms of cost. In the meantime, existing PCMs require more attention in the following aspects:(a)Due to a huge number of mega projects and a strong demand in the Saudi housing and entertainment industries, the growth of the construction industry in the next years will become more significant. The adoption of new creative technologies and processes, such as integrating PCMs into building walls in the building sector, has therefore become an essential need to increase highly efficient structural operations.(b)Novel and efficient encapsulation techniques for PCMs, such as UV-based encapsulation process of hydrated salts and paraffin waxes require further investment in terms of their stability over time in real-life applications.(c)More information on the thermal and physical properties of PCMs is expected, especially in the area of their thermal stability during prolonged freezing/cooling cycles. The dispersion of nanoparticles can have a positive effect on thermal stability, which has been underestimated in previous research.(d)The corrosive properties of PCMs is another area where more research is planned. If PCMs lose their anti-corrosive properties with time, this could prove disastrous and appropriate measures should be taken beforehand.(e)Finally, yet importantly, PCMs are not a substitute for conventional insulation materials; therefore, possibilities to combine the use of PCMs with conventional insulation materials should be investigated. One solution for this may be the direct integration of PCMs into insulation materials.

## Figures and Tables

**Figure 1 materials-14-05328-f001:**
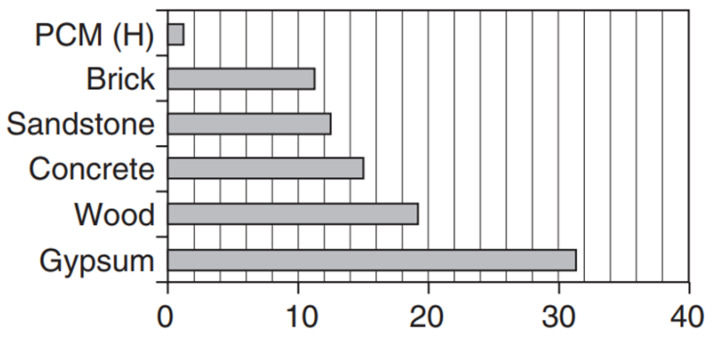
The thickness of PCMs required, compared to conventional thermal masses (e.g., gypsum, wood, concrete, sandstone and brick), for building envelope applications. This figure was redrawn based on [3].

**Figure 2 materials-14-05328-f002:**
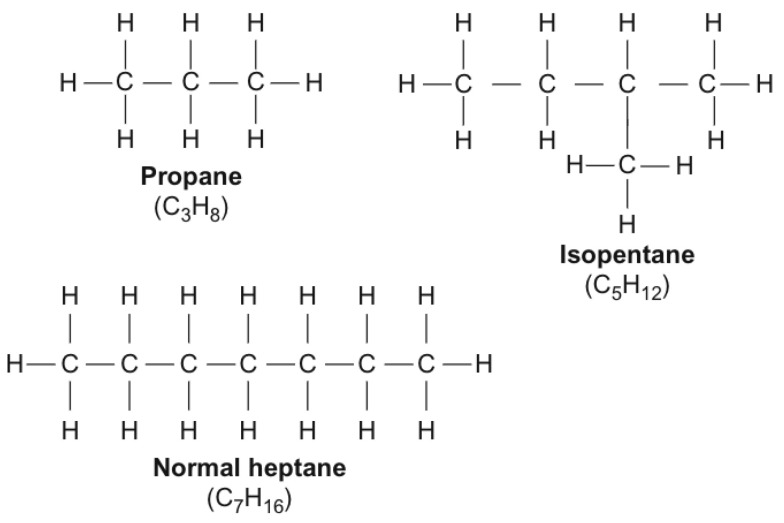
Examples of paraffins. This figure was redrawn based on [24].

**Figure 3 materials-14-05328-f003:**
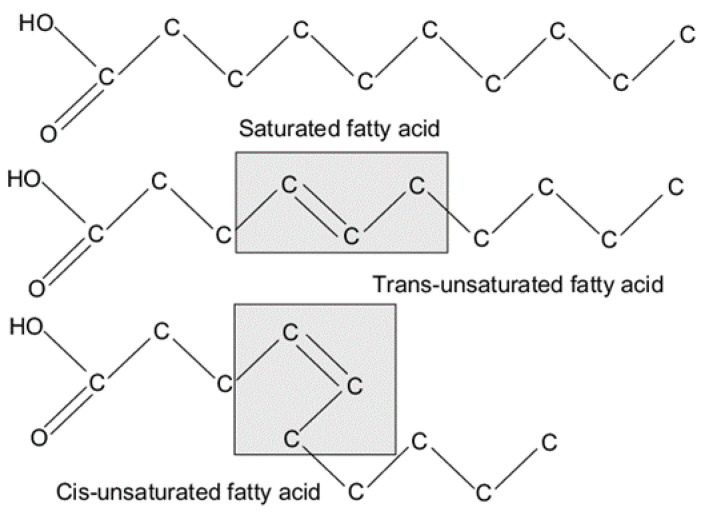
Molecular structure of different types of fatty acids. This figure was redrawn based on [43].

**Figure 4 materials-14-05328-f004:**
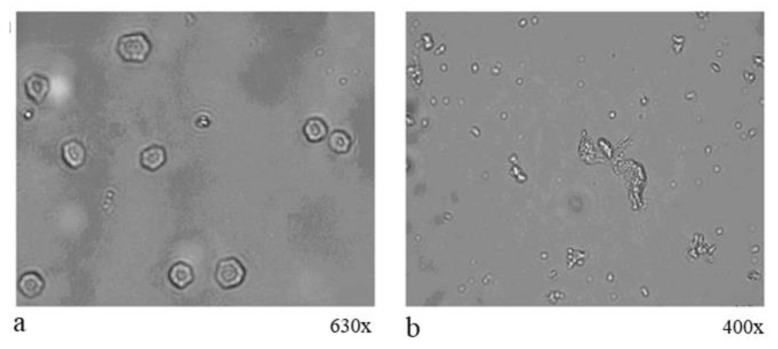
Optical micrographs exhibiting BS micro-encapsulation in polymers with magnification ×630 (**a**), and magnification ×400 (**b**) [81].

**Figure 5 materials-14-05328-f005:**
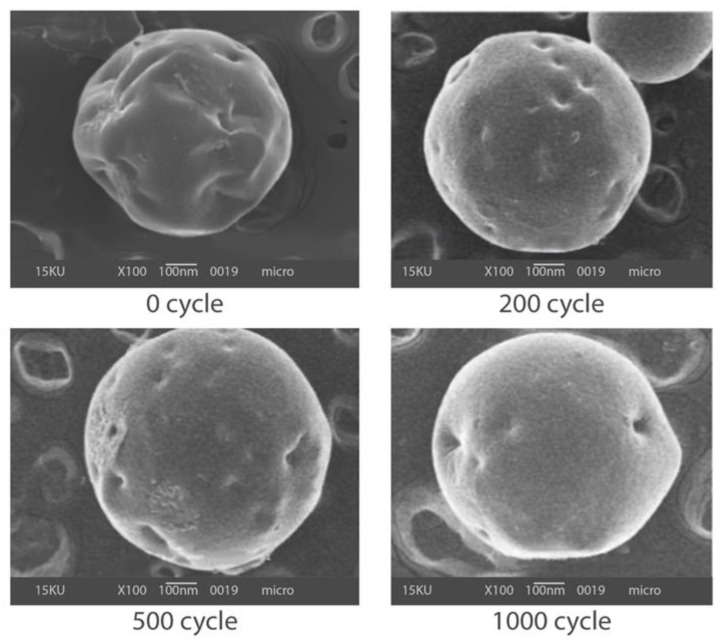
SEM images of paraffin particles after micro-encapsulation and subjected to thermal cycles [74].

**Figure 6 materials-14-05328-f006:**
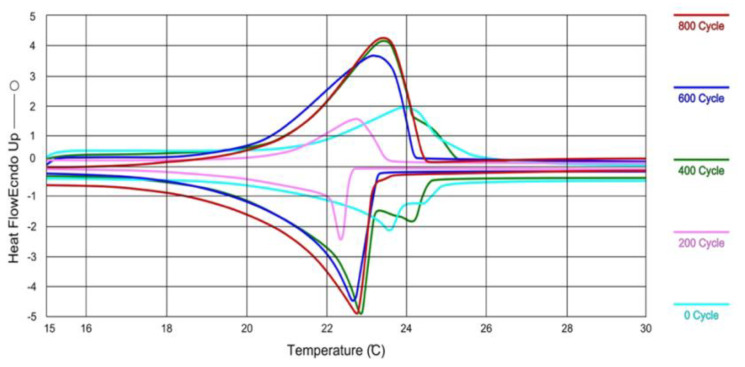
Comparison of the DSC curves of micro-encapsulated BS PCMs subjected to 0, 200, 400, 600 and 800 thermal cycles [75].

**Figure 7 materials-14-05328-f007:**
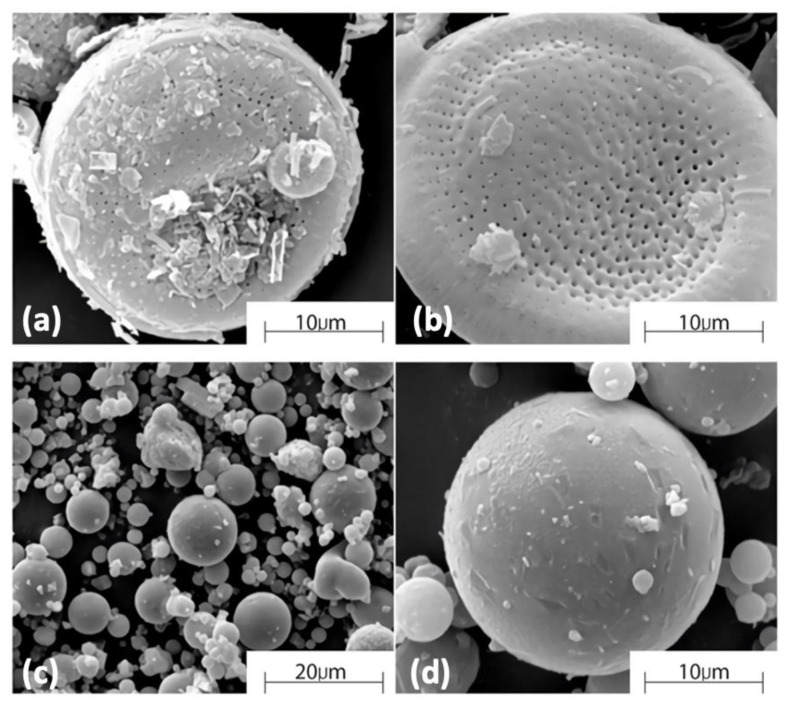
SEM micrographs of diatomite/paraffin composite: (**a**) diatomite in as-received condition, (**b**) diatomite after the calcination process, (**c**) composite structure of paraffin/diatomite and (**d**) higher magnification view of single composite sphere [76].

**Figure 8 materials-14-05328-f008:**
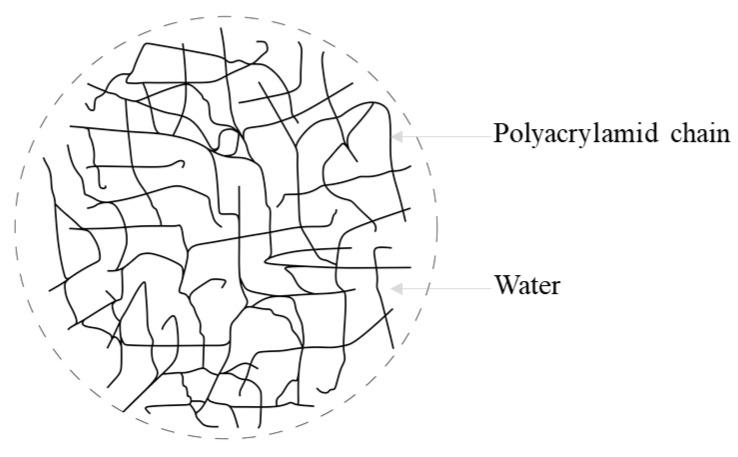
Micro-encapsulation of hydrated salt PCMs in polyacrylamide film [104].

**Figure 9 materials-14-05328-f009:**
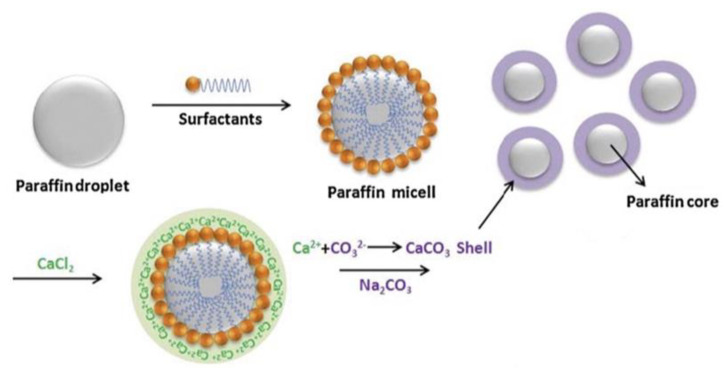
Schematic of paraffin micro-encapsulation in CaCO_3_ shell [78].

**Figure 10 materials-14-05328-f010:**
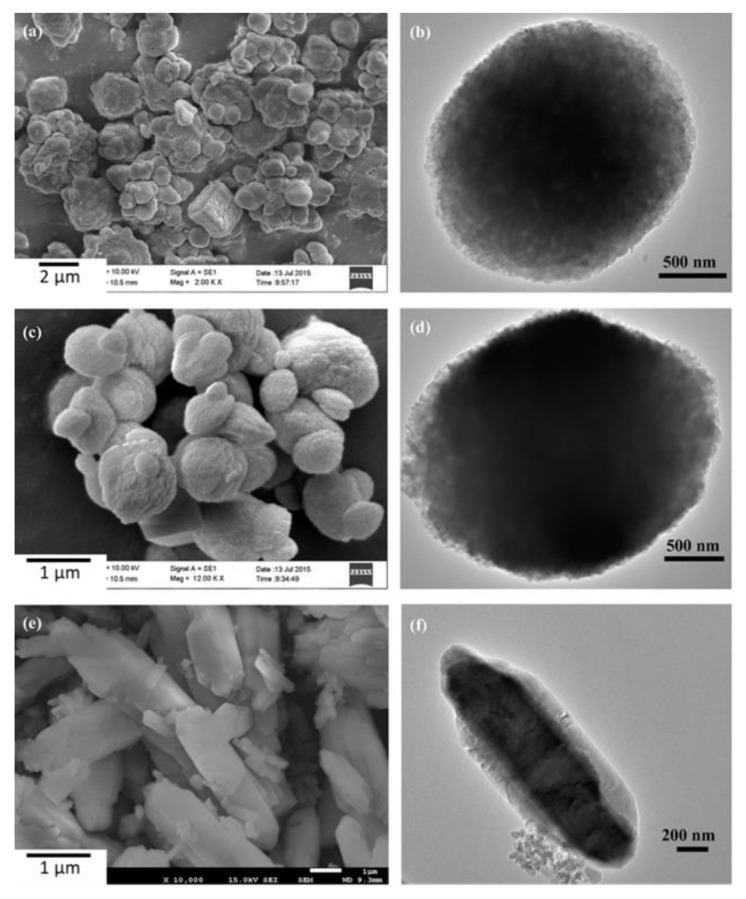
Morphology of micro-encapsulated paraffin in CaCO_3_ shell: SEM images (**a**,**c**,**e**) and TEM images (**b**,**d**,**f**) [78].

**Figure 11 materials-14-05328-f011:**
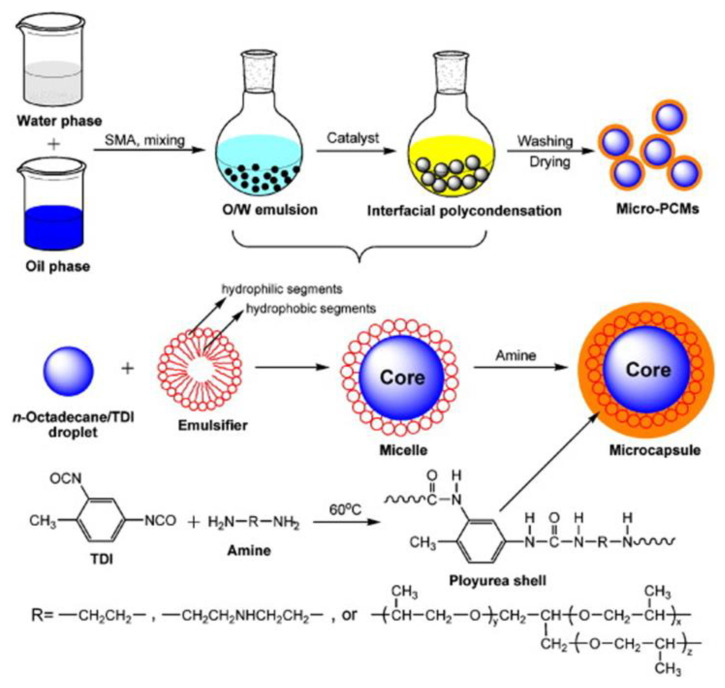
Schematic of microencapsulation process of *n*-octadecane with poly-urea shells [106].

**Figure 12 materials-14-05328-f012:**
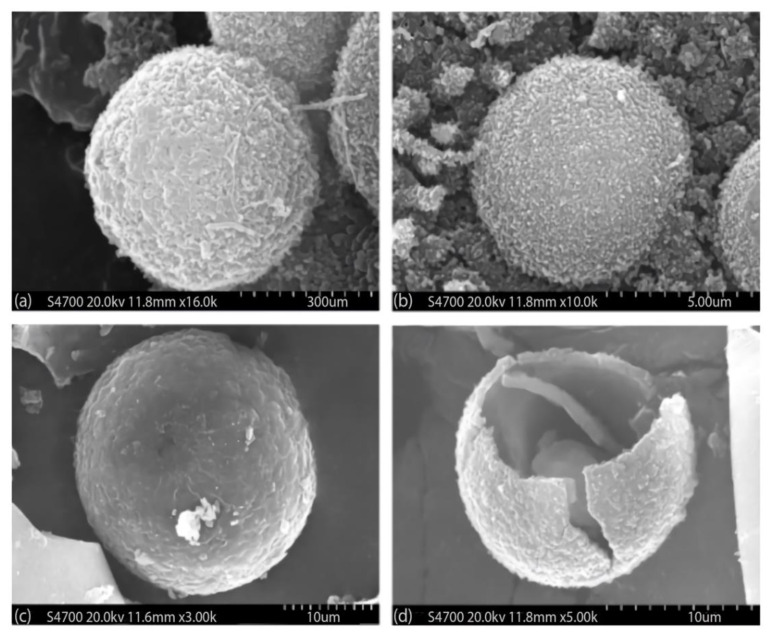
SEM images of micro-encapsulated PCMs using different monomers: (**a**) ethylene diamine (EDA), (**b**) toluene-2,4-diisocyanate (TDI), (**c**) Jeffammine and (**d**) a cracked microcapsule [106].

**Figure 13 materials-14-05328-f013:**
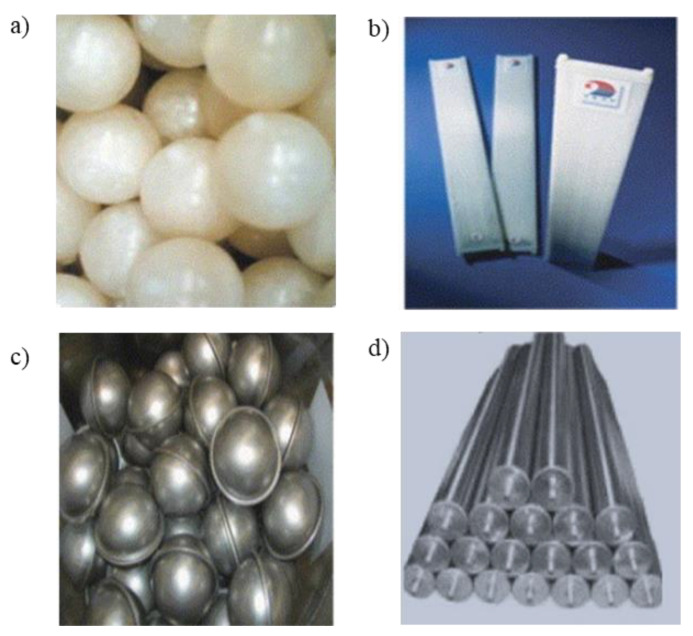
Commercially manufactured macro-capsules to store PCMs: (**a**) Polyolefin spherical balls, (**b**) Polypropylene flat panel, (**c**) stainless ball capsules, and (**d**) modules beam [36].

**Figure 14 materials-14-05328-f014:**
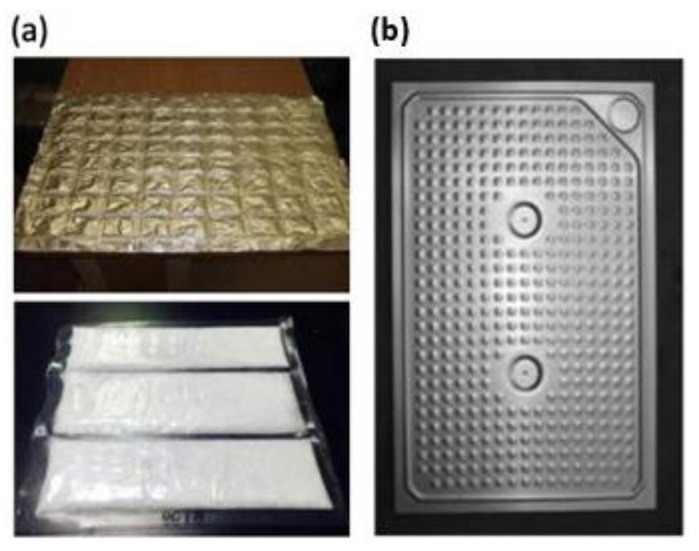
Macro-encapsulation of PCMs in (**a**) pouches and (**b**) flat panels [107].

**Figure 15 materials-14-05328-f015:**
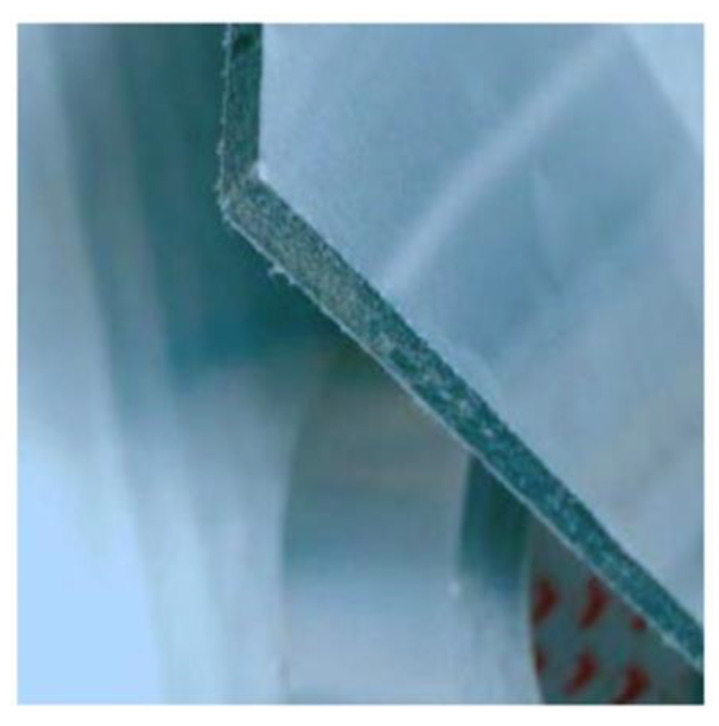
Energain^®^ PCM panel wrapped and sealed in foil tape [141].

**Figure 16 materials-14-05328-f016:**
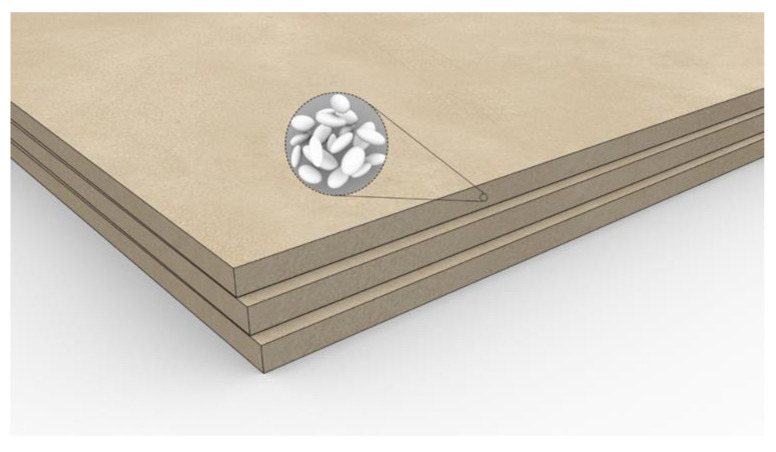
Micronal^®^ PCM-integrated gypsum wallboard. This figure was redrawn based on [140].

**Figure 17 materials-14-05328-f017:**
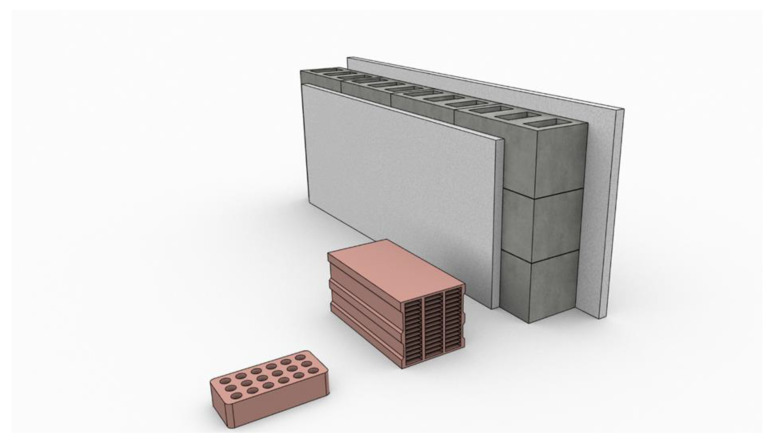
Hollow brick, alveolar brick and wall structure by using PCMs.

**Figure 18 materials-14-05328-f018:**
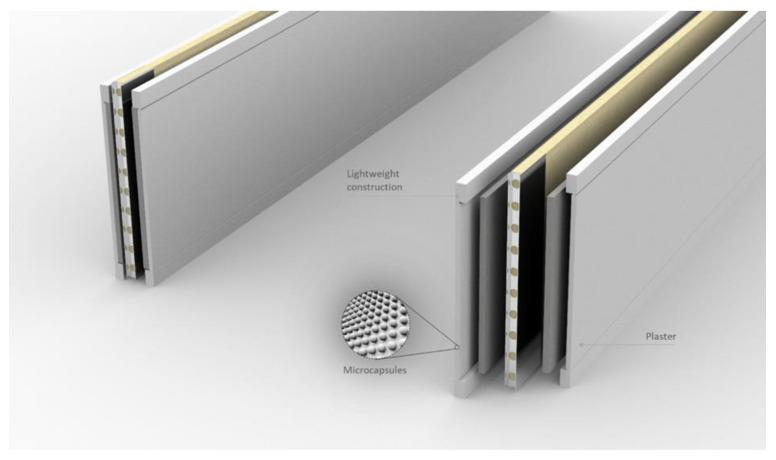
PCM usage as interior plaster on hollow building blocks.

**Figure 19 materials-14-05328-f019:**
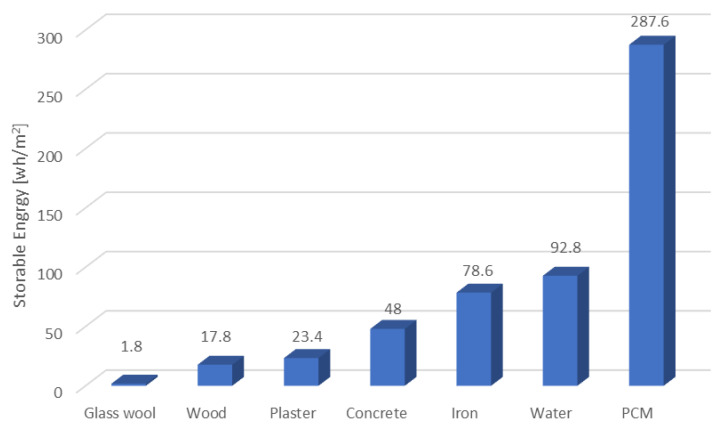
Heat storage capacity of different building materials between 18 °C and 26 °C for a period of 24 h. This figure was redrawn based on [111].

**Table 1 materials-14-05328-t001:** Classification of PCMs, together with their common properties [10,12,13,14,15,16,17,18,19,20,21,22,23].

PCM Properties	PCM Type		
	Organic	Inorganic	Eutectics
Examples of PCMs	Paraffin, non-paraffin compounds, fatty acids, acetamide, butyl stearate	CaCl_2_.6H_2_O, Na_2_SO_4_.10H_2_O, Na_2_CO_3_.10H_2_O	Octadecane + heneicosane, Octadecane + docosane, 34% C_14_H_28_O_2_ + 66% C_10_H_20_O_2_
Melting temperature (°C)	19–32	29–36	25–27
Thermal conductivity (w/m.k)	0.15–0.21	0.54–1.09	-
Heat of fusion (J/kg K)	140–236	105–192	136–203
Density (kg/m^3^)	756–815	1600–1800	-

**Table 2 materials-14-05328-t002:** Thermo-physical properties of different paraffin waxes.

PCMs	T_m_ (°C)	K (W/m.K)	H (kJ/kg)	ρ (kg/m^3^)	Ref.
Paraffin RT-18	15.0–19.0	0.20	134.0	0.756	[29]
Paraffin RT-27	28.0	0.20	179.0	0.800	[30]
n-Octadecane	28.0	0.20	179.0	0.750	[31]
	28.0	0.15	200.0	-	[32]
n-Heptadecane	19.0	0.21	240.0	760.0	[32]
	22.0	-	244.0	780.0	[33]
Hexadecane	18.0	0.17–0.26	236.0	780.0	[33]
Polyethylene glycol	21.0–25.0	-	148.0	1128.0	[34]
Paraffin C_13_–C_24_	22.0–24.0	0.21	189.0	0.760	[32,35]
n-Octadecane + n-Heneicosane	26.0	-	173.9	-	[36]
Paraffin C_16_–C_18_	20.0–22.0	-	152.0	-	[11,35,37]
Paraffin C17	21.70	-	213.0	0.817	[38]
Paraffin C_18_	28.0	0.15	244.0	0.774	[11,35,36,37,38]

Legend: H: heat of fusion. k: thermal conductivity. Tm: melting temperature. ρ: density.

**Table 3 materials-14-05328-t003:** Thermo-physical properties of different fatty acids.

PCMs	T_m_(°C)	K (W/m.K)	H (kJ/kg)	ρ (kg/m^3^)	Ref.
Lactic acid	26.0	-	184.0	-	[38]
Capric acid	30.2	0.20	142.7	-	[44]
	30.0–30.2	0.2	143.0	815.0	[44,45]
Stearic acid	52.0	0.29	169.0	965.0	[46]
Palmitic acid	62.4	-	183.2	-	[47]
Myristic acid	52.7	-	198.4	-	[47]
Capric acid + Lauric acid	21.0	-	143.0	-	[11]
	19.0	-	132.0	-	[36]
	20.39	-	144.2	-	[48]
Capric acid + Stearic acid	26.8	-	160.0	-	[36]
Capric acid + Palmitic acid	22.1	-	153.0	-	[36]
	26.2	2.20	177.0	784.0	[49]
Myristic acid + Capric acid	21.4	-	152.0	-	[36]
Capric acid + 1-dodecanol	26.5	-	-	-	[44]
	27.0	-	126.9	817.0	[44]
	26.5	0.2	126.9	817.0	[50]
Methyl Palmitate + Methyl Stearate	23.0–26.5	-	180.0	817.0	[51]
Dodecanoic acid	42.5	0.148	182.0	873.0	[52]
Glycerin	18.0	-	199.0	-	[29]
Capric acid (75.2%) + Palmitic acid (24.8%)	22.1	-	153.0	-	[53]
Capric acid (75%) + Palmitic acid (25%)	17.7–22.8	-	189.0–191.0	-	[54]
Capric acid (86.6%) + Stearic acid (13.4%)	26.8	-	160.0	-	[53]
Capric acid (61.5%) + Lauric acid (38.5%)	19.1	-	132.0	-	[53]
Lauric acid (55.8%) + Myristic acid (32.8%) + Stearic acid (11.4%)	29.29	-	28.38	-	[55]
Expanded Graphite (Lauric acid + Myristic acid + Stearic acid)	29.05	-	137.0	-	[55]
Capric acid + Palmitic acid + Stearic acid	19.93	-	129.5	-	[56]
Myristic acid + Palmitic acid + Stearic acid	41.72	-	163.5	-	[56]
Expanded Graphite (Myristic acid + Palmitic acid + Stearic acid)	41.64	-	153.5	-	[56]
Expanded Perlite (Capric acid + myristic acid)	21.7	-	85.4	-	[57]
Activated Carbon (Lauric acid)	44.1	-	65.14	-	[58]
Expanded Graphite (Stearic acid)	53.12	-	155.5	-	[59]
Diatomite (Capric acid + Lauric acid)	16.7	-	66.8	-	[60]
Activated Montmorillonite (Stearic acid)	59.9	-	84.4	-	[61]
Expanded Graphite (Palmitic acid)	60.9	-	148.4	-	[62]

Legend: H: heat of fusion. k: thermal conductivity. Tm: melting temperature. ρ: density.

**Table 4 materials-14-05328-t004:** Thermo-physical properties of different hydrated salts.

PCMs	T_m_ (°C)	K (W/m.K)	H (kJ/kg)	ρ (kg/m^3^)	Ref.
Hydrated salts	29.0	1.0	175.0	1490.0	[64]
	31.4	-	150.0	-	[65]
	25.0–34.0	-	140.0	-	[66]
	26.0	0.60	180.0	1380	[30]
Eutectic salt	32.0	-	216.0	-	[67]
Sodium Sulfate Decahydrate	32.50	0.60	180.0	1600	[68]
FeBr_3_·6H_2_O	21.0	-	105.0	-	[38]
Mn(NO_3_)·6H_2_O	25.5	-	126.0	1738	[11,35,38]
	25.8	-	125.9	1728	[46]
Mn(NO_3_)_2_·6H_2_O + MnC_l2_·4H_2_O	27.0	0.60	125.9	1700	[69]
CaC_l2_·6H_2_O	29.0	0.54	187.49	560	[70]
	29.9	0.53	187.0	1710	[71]
Sodium thiosulfate pentahydrate	40.0–48.0	-	210.0	-	[72]
Sodium acetate trihydrate	58.0	1.10	264.0	1280	[73]
Na_2_SO_4_·10H_2_O-Na_2_CO_3_·10H_2_O	32.34	-	-	-	[74]
Hexahydrate (CaCl_2_.6H_2_O)	30.0	1.10	170.0	1560.0	[75]
Decahydrate (Na_2_SO_4_.10H_2_O)	37.7	-	-	131.7	[76]
KF_4_H_2_O	18.50	-	231.0	1447.0	[77]
Na_2_SO_4_·10H_2_O	21.0	0.55	198.0	1480.0	[78]
Calcium chloride	29.8	0.56	191.0	1710	[79]

Legend: H: heat of fusion. k: thermal conductivity. Tm: melting temperature. ρ: density.

**Table 5 materials-14-05328-t005:** Thermo-physical properties of different butyl stearate PCMs.

PCMs	T_m_ (°C)	K (W/m.K)	H (kJ/kg)	ρ (kg/m^3^)	Ref.
Butyl stearate	16.0–20.8	0.21	700.0	900	[75]
	18.0	-	30.0	-	[75]
	19.0	-	140.0	760	[53]
	18.0–23.0	0.21	123.0–200.0	-	[53]
BS/MMT	25.30	-	41.81	-	[85]
Butyl Stearate & Butyl Palmitate (49/48)	17.0–20.0	-	137.8	-	[86]
Butyl Stearate & Butyl Palmitate (50/48)	15.0–25.0	-	101.0	-	[87]
CH_3_(CH_2_)_16_COO(CH_2_)_3_CH_3_	19.0	-	140.0	-	[88]
Butyl stearate (50%) and Butyl palmitate (48%)	16.0–21.0	-	-	-	[52,89]
Butyl stearate (48%) and Butyl palmitate (49%)	17.0–19.3	-	-	-	[90]

Legend: H: heat of fusion. k: thermal conductivity. Tm: melting temperature. ρ: density.

**Table 6 materials-14-05328-t006:** Thermo-physical properties of micro-/macro-encapsulations suitable for wall construction applications available in the literature.

PCMs Core	Shell Materials	Encapsulation Efficiency (%)	Size of Particles (µm)	Maximum Latent Heat (J/g)	Melting Temperature (°C)	Ref.
N–octadecane	Polyurethane	93.4–94.7	5.0–10.0	110.4	28.0	[108]
	Poly(methyl methacrylate-co-methacrylic acid) copolymer	12.0–21.0	1.60–1.68	93.0	29.0–32.9	[109]
	poly(n-butyl methacrylate) & poly(n-butyl acrylate)	47.7–55.6	2.0–5.0	112.0	29.10	[110]
	Melamine Formaldehyde co-polymer	-	34.0	183.2	28.14	[111]
	Silk fibroin	22.6–46.7	4.0–5.0	-	24.99	[112]
	SiO_2_/PMMA	19.9–66.4	5.0–15.0	-	21.5–26.3	[113]
	TiO_2_/PMMA	26.8–82.8	3.0–16.0	100.0	28.0–31.0	[114]
	Resorcinol-modified melamine	44.0–69.0	5.0–20.0	146.5	26.5–28.4	[115]
	SiO_2_/TiC(PMMA)	78.0	-	-	17.2–19.4	[116]
	Poly(MPS-VTMS)	58.7–76.0	-	166.7	17.4–18.2	[117]
	Octadecylamine-grafted	<88.0	-	202.5	27.4–27.5	[118]
	SiO_2_	33.6	8.0	210.0	23.3–28.4	[119]
	Calcium Carbonate (CaCO_3_)	22.4–40.4	5.0	-	28.1–29.2	[120]
	TiO_2_	74.3–81.0	2.0–5.0	42.6	25.6–26.1	[112]
N-nonadecane	Poly(methyl methacrylate)	60.30	0.1–35.0	139.2	31.2	[121]
	CaCO_3_	40.04	5.0	84.40	29.2	[120]
N-heptadecane	Poly(styrene)	63.3	1.0–20.0	136.9	21.5	[121]
	Starch	49.0–78.3	30.0–175.0	187.3	23.1–24.2	[122]
N-eicosane	Polysiloxane	-	5.0–22.0	240.0	35.0–39.0	[123]
	Crystalline TiO_2_	49.9–77.8	1.5–2.0	97.60–195.6	41.5–43.88	[124]
N-octadecane, N-eicosane, and N-nonadecane	Melamine-Urea-Formaldehyde	-	0.30–6.40	165	36.40	[125]
N-octadecane (paraffin wax)	Melamine formalde hyderesin	92.0	2.0–5.0	214.6	28.41	[115]
Paraffin (MPCM24D)	melamine-formaldehyde polymer	-	10.0–30.0	154.0	21.9	[125]
Paraffin wax	Polystyrene	75.60	-	153.50	-	[126]
	Amphiphilic polymer (PE-EVA-PCM)	-	-	98.1	28.4	[127]
	Hydrophobic polymer (St-DVB-PCM)	-	-	96.1	24.2	[127]
Paraffin eutectic	poly(methyl methacrylate)	50.20–65.40	0.01–100	276.41	36.17	[128]
Butyl stearate	Polyurethane	-	10.0–35.0	81.20	22.30	[129]
Paraffin	Melamine-formaldehyde	80.0	5.8–339.0	147.9	21.0–24.0	[130]
	Melamine-formaldehyde	80	10.4–55.2	147.9	22.5	[131]
	Poly-methyl-methacrylate	-	50.0–300.0	100.0	23.0	[66]
	Ethylvinylacetate and polyethylene	-	3.0–10.0	100.0	27.0	[132]
	polymethylmethacrylate	-	6	135.0	23.0	[133]
Graphite-modified MPCM	Polycarboxylate		3.0–3000.0			[133]

**Table 7 materials-14-05328-t007:** Summary of different experimental studies in 2020 from the literature on wall applications.

PCMs	Application in Building Walls	Improvements	Ref.
Paraffin wax	Two-wall models were tested; walls were insulated with wood to achieve one-dimensional heat transfer through the walls.	The PCM layer provides a more uniform temperature rise near the heat source of the building wall, reducing the heat flow through the downstream parts	[106]
OM37 PCM	The experimental setup consists of a PCM-free reference concrete cubicle and a PCM-macro-encapsulated experimental concrete cubicle. The cubicles were both installed in an open area and are directly exposed to solar energy	All four walls of the cubicle show more or less the same temperature profile over 24 h. The test walls of the cubicle interior surface temperature remain below the daytime temperature compared to the reference cubicle walls	[107]
OM35 and Eicosane	Brick	For the double PCM layer within the brick, there is a heat loss of approx. 9.5 °C and for a single-layer PCM brick a temperature reduction of 6 °C. The reduction of heat gain for double-layer PCM bricks is observed up to 60% during the day, and about 40% for single-layer PCM bricks. However, this is not the case when the heat is denied during the night by PCMs. The use of small particles within the PCM encapsulation has a detrimental effect as the heat transfer increases sharply during the day. In these bricks, the internal brick temperature increases more than normal bricks during the night and the PCM was not properly dissipated. Although heat dissipation during the night is ensured by some secondary means other than using the fin configuration, it may not be a viable choice to increase the PCM thickness for cooling a building space	[108]

**Table 8 materials-14-05328-t008:** Summary of different experimental studies in 2019 from the literature on wall applications.

PCMs	Application in Building Walls	Improvements	Ref.
Paraffin wax	Blocks made from foamed cement with paraffin	(1) Additional paraffin can make the foamed cement block more effective in storing thermal energy. (2) The method of preparation and absorption is a physical process. During the process shifts in the thermal storage of molded cement, the thermal storage properties and the chemical properties of the paraffin are obtained. (3) The thermal conductivity of the pure cement block is similar to that of the foamed cement block with 20 percent to 25 percent composite PCM. Thirty percent is the optimum mass ratio and displays the best thermal inertness for foamed cement blocks with composite PCM. They effectively slow down the temperature rise and minimize temperature fluctuations, but higher temperatures occur after process changes.	[113]
Paraffin wax (SSPCM)	SSPCM boards with EPS board	(1) It was found that the PCM wallboard keeps more heat out of the room air during the day and releases it at night through the heat flux of the wall. (2) In summer, the room temperature in the PCM area was 1.9 °C below the maximum temperature and 0.6 °C below the average temperature in the reference room. (3) The inside environment temperature in the PCM room in the summer was 1.9 °C below the maximum temperature and 0.6 °C below the reference testing room average temperature. (4) In winter, the indoor air temperature in the PCM space was 1.3 °C lower at altitude and 0.1 °C higher on average than in the comparison room. (5) It was found that the PCM wallboard stores more heat from the indoor air during the day and releases it from the heat flux of the wall surface at night, resulting in less heat being released to the interior during the day. In winter, the case is reversed and it was found that the PCM wallboard stores more heat from the outside air during the day and releases it at night, resulting in more heat being transferred to the inside of the wall at night	[109]
Fatty acids	G/C board consisting of inorganic materials as well as gypsum and cement.	They verified that the heat storage G/C board applied with PCM was effective in reducing the energy inside buildings.	[110]
PH-31 Paraffin wax	Gypsum mortar	The thermal conductivity of conventional gypsum wallboard increases gradually with an increase in temperature. The integration of the micro-PCM into the gypsum would minimize the thermal conductivity on the MPCM-containing wallboard by reducing its density and lowering its conductivity below the density of the gypsum wallboard. The composite wall panel has a much higher apparent specific heat capacity than a gypsum wallboard (2.71 times in the 26 °C to 32 °C temperature range).	[116]
The microcapsules with a paraffin core and a melamine-formaldehyde polymer shell	Geo-polymer concrete walls	The annual energy savings from using walls with 15 cm GPC-MPCM (5.2 wt.%), a 5-cm PCM layer, and 5 cm insulation was approximately 28% compared to the comparison by maintaining the indoor temperature of 19 °C–21 °C. The PCM layer performed better when positioned closer to the outdoor climate.	[139]

**Table 9 materials-14-05328-t009:** Summary of different experimental studies in 2018 from the literature on wall applications.

PCMs	Application in Building Walls	Improvements	Ref.
Paraffin wax	Brick wall	(1) PCMs in structures have been used as thermal insulating products and increased thermal comfort. (2) PCM decreases the temperature of the indoor environment and lowers the cooling charge.	[117]
Calcium chloride hexahydrate (CaC1_2_ · 6H_2_O)	Concrete walls contain glass windows with different positions (inside, center, outside)	The use of the PCM in the walls greatly reduced the heat transfer rate and the average internal surface temperature during the working hours. Liquid fraction analysis was performed to determine the effective fraction of the total PCM that could be used to build the PCM layer in the most efficient manner. The heat reduction ratio was used as a measure of the performance of the PCM substrate, varying the shape, thickness, and PCM locations	[118]
(PCM) used is a mixture of ethylene-based polymer (40%) and paraffin wax PCM (60%).	Aluminum sheet	A considerable reduction of thermal losses around the walls. These losses are decreased by 50 percent on average. When placed on a ceiling the PCM is more effective.	[119]
	Wallboard	PCM wall panel has been called low convection heat wallboard, low relative thermal conductivity or low heat ratio wallboard.	[122]
BASF Micronal^®^ DS 5038X	Precast concrete	Not only because of the direct effect of replacing the heavier mixture fraction with the lighter one, but also because the compressed air increases with the implementation of increasing PCM volumes, a statistically important reduction in density is produced by the concentrations of microencapsulated PCM.	[123]

**Table 10 materials-14-05328-t010:** Summary of different experimental studies in 2017 from the literature on wall applications.

PCMs	Application in Building Walls	Improvements	Ref.
Paraffin wax	Cement mortar	Only because of the small and fixed-phase temperature range, the composite PCM wall can work efficiently. The advantages of the PCM insert should be optimized by selecting a suitable melting point under appropriate test conditions.	[124]
Paraffin wax	Brick holes	The following was observed: the n-octadecane PCM decreases the cooling rate for the first day in the evaluation by more than 50% compared to the other PCM blocks, with n-eicosane causing a reduction of about 40% and the P116 causing a reduction of about 30%.	[126]

**Table 11 materials-14-05328-t011:** Thermal properties of commercial PCMs used in different wall components, as discussed in the literature.

PCMs	Location	T_m_ (°C)	K (W mK^−1^)	H (kJ kg^−1^)	ρ (kg m^−3^)	Type of Wall	Method	PCM Supplier	Ref.
RT 10	Guimarães, Portuga	10.0	-	150.0	880.0	Plaster mortar for exterior wall	Experimental and numerical	Rubitherm GmbH	[136]
RT 18	Shanghai, China	17.0–19.0	0.20	225.0	770.0	Wallboard	Experimental and numerical	Rubitherm GmbH	[136]
	Various Portuguese cities	15.0–19.0	0.20	134.0	0.756	Masonry brick	Experimental study	Rubitherm GmbH	[29]
RT 20	-	18.0–22.0	0.20	172.0	810.0	Gypsum wallboards	Computer simulation	Rubitherm GmbH	[136]
	China	23.2	-	134.1	-	Gypsum wallboards	Gypsum wallboards	Rubitherm GmbH	[85]
RT 20/MMT	China	24.2	-	53.60	-	Gypsum wallboards	Gypsum wallboards	Rubitherm GmbH	[85]
RT 21	Puigverd de Lleida, Spain	21.0	0.20	134.0	0.770	Prefabricated Slab Concrete	Experimental and numerical	Rubitherm GmbH	[133]
	Puigverd de Lleida, Spain	21.0–22.0	-	134.0	-	Prefabricated Slab Concrete	Experimental study	Rubitherm GmbH	[136]
	Sydney, Australia	21.0	0.20	-	880.0	Trombe walls	Experimental study	Rubitherm GmbH	[133]
RT 25	Raebareli Uttar Pradesh & Bhopal, India	26.6	0.18	232.0	749.0	Building bricks	Numerical Study	Rubitherm GmbH	[144]
RT 27	Spain	27.0	0.12	100.0	-	Geopolymer concrete & Cement concrete	Experimental study	Rubitherm GmbH	[136]
	Lawrence, Kansas, USA	27.0	0.20	179.0	760.0	Gypsum wallboard	Experimental and numerical	Rubitherm GmbH	[144]
	Sydney, Australia	25.0–28.0	0.20	-	880.0	Trombe walls	Experimental study	Rubitherm GmbH	[136]
RT 31	Sydney, Australia	27.0–31.0	0.20	-	880.0	Trombe walls	Experimental study	Rubitherm GmbH	[133]
RT 42	-	38.0–43.0	0.20	174.0	760.0	Solar chimney	Experimental study	Rubitherm GmbH	[144]
	Sydney, Australia	38.0–43.0	0.20	-	880.0	Trombe walls	Experimental study	Rubitherm GmbH	[136]
RT 22 HC	Ljubljana, Slovenia	21.0–22.0	0.18	134.0	677.0	Wallboard	Experimental and numerical	Rubitherm GmbH	[133]
RT 28 HC	Coimbra, Portugal	28.0	0.20	245.0	756.0	Trombe wall	Experimental study	Rubitherm GmbH	[144]
	Coimbra, Portugal	27.55	0.20	258.1	-	Trombe wall	Experimental study	Rubitherm GmbH	[133]
GR 35	Erzurum, Turkey	13.0–41.0	-	41.0	-	Trombe wall	Experimental study	Rubitherm GmbH	[144]
GR 41	Erzurum, Turkey	13.0–51.0	-	55.0	-	Trombe wall	Experimental study	Rubitherm GmbH	[144]
SP29	Shanghai, China	28.0–30.0	0.60	190.0	1520.0	Wallboard	Experimental and numerical	Rubitherm GmbH	[136]
PEG-600	France	21.0–25.0	-	148.0	1128.0	PVC panel	Experimental and numerical	-	[34]
	Slovenia	18.0–27.0	0.20	330.0	-	Wallboard	Simulation	Rubitherm GmbH	[144]
Micronal^®^ PCM	Puigverd de Lleida, Spain	26.0	-	110.0	-	Concrete walls	Experimental study	BASF	[144]
ENERGAIN	Lyon, France	23.5	0.22	107.5	900	Wallboard	Experimental study	Dupont de Nemours	[85]
Methyl Palmitate (Emery 2216)	Canada	24.0–28.0	-	192.0	-	Wallboard	Experimental study	Henkel	[51]
Methyl Stearate (Emery 2218)	Canada	33.0–36.0	-	196.0	-	Wallboard	Experimental study	Henkel	[51]
MC-24	Guimarães, Portuga	24.0	-	162.40	-	Plaster mortar for exterior wall	Experimental and numerical	DEVAN (MC series)	[136]
MC-28	Guimarães, Portuga	28.0	-	170.0	350	Plaster mortar for exterior wall	Experimental and numerical	DEVAN (MC series)	[136]
BSF26	Guimarães, Portuga	26.0	-	110.0	-	Plaster mortar for exterior wall	Experimental and numerical	BASF	[136]
GH 20	Shanghai, China	25.0–25.4	0.82	33.25	1150.0	Shape-stabilized mortar bricks	Experimental study	-	[133]
Energain^®^	Coimbra, Portugal	18.0–26.0	855	70.0	2500.0	Drywalls	Experimental and numerical	DuPont™	[133]
	Bayern, Germany	-	0.18–0.23	-	855.0	Insulation in lightweight walls	Experimental and numerical		[144]
Q25- BioPCM™	Melbourne Australia	28.2	0.20	242.0	235.0	Internal and external building walls	Experimental and simulation	Phase Change Energy Solutions, Inc.	[133]
Micronal^®^ DS (5001-X)	Coimbra, Portugal	25.67	0.15	111.3	-	Trombe wall	Experimental study	BASF	[144]
	Spain	26.0	-	179.0	-	Gypsum and Portland cement	Experimental study	BASF	[133]
PEG-E600	Madurai, India	25.0–31.0	-	180.0	1126	Inside hollow-brick façades	Experimental study	BASF	[35,36]
L-30	Würzburg, Germany	30.0	1.02	270.0	-	Building walls	Experimental study	Rubitherm GmbH	[133]
S-27	Würzburg, Germany	27.0	0.48	190.0	-	Building walls	Experimental study	Rubitherm GmbH	[133]
PCM-HDPE	USA	16.6–26.5	-	116.7	505.3	Insulation in wall cavities	Experimental and numerical	PCM Products Ltd.	[63]
GKB^®^	Athens, Greece	16.0–26.0	0.27	-	787.0	Plasterboards	Experimental study	Knauf	[133]
Micronal-T23^®^	Southern Italy	19.0–25.5	0.18–0.22	-	545.0	Wallboard	Experimental study	BASF	[136]
Micronal^®^	Chile	25.0	0.23	-	800	Gypsum board	Simulation study	Knauf	[144]
PT-20	Auckland, New Zealand	20.0	-	180.0	-	Gypsum board	Experimental study	PureTem	[133]
Emerest 2325	Montreal, Canada	17.0–20.0	-	137.8	-	wallboard	Experimental study	Henkel	[86]
Emerest 2326	Montreal, Canada	15.0–25.0	-	101.0	-	Autoclaved block	Experimental study	Henkel	[87]
MF/PCM24	Oslo, Norway	21.8	0.02–0.10	154.0	-	Insulation material & Geopolymer concrete	Experimental study	Microtek	[139]
Micronal DS-5008X	Lisbon & Porto, Portugal	23.0	0.30	100.0–110.0	300.0	Plastering Mortar	Numerical study	BASF	[88]
Micronal DS-5040X	Australia	23.0	-	100.0	250.0–350.0	Cement-based materials	Experimental study	BASF	[133]
Micronal DS-5008X	Portugal	23.0	-	135.0		Cement, lime, and gypsum mortars	Experimental study	BASF	[133]
Micronal^TM^ ThermalCORE	United States	23.0	0.20	24.2	800.0	Drywall	Experimental study	ThermalCore Inc	[133]
M-51	United States	23.0	0.15	230.0	860.0	Plastic foil inside wallboard	Experimental study	Bio-PCM	[133]
InfiniteR^TM^	United States	23.0	0.54	200.0	1810.0	PE foil bags inside wallboard	Experimental study	Infinite Business Solutions	[133]
M182Q25	Canada	25.0	0.15–2.5	210.0–250.0	-	Building walls and concrete slab	Experimental study	Bio-PCM	[144]
M51Q25	Canada	25.0	0.15–2.5	210.0–250.0	-	Building walls and concrete slab	Experimental study	Bio-PCM	[136]
PureTemp-20	Los Angeles, USA	10.0–28.0	0.21	100.0–400.0	860.0	Concrete composites walls	Experimental study	Entropy Solution Inc.	[144]
SP 25-A8	Spain	26.0	0.60	180.0	1380	Hollow bricks	Experimental study	Rubitherm GmbH	[30]

## Data Availability

Data will be available on suitable demand.

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
