# Peer review of "Potential Phase Change Materials in Building Wall Construction—A Review"

_materials, 2021, doi:10.3390/ma14185328_

Round 1

Reviewer 1 Report

This manuscript from Kurdi et al. shows their review of the phase change materials’ potential applications in the building wall construction. This is an important topic as the energy can be used much wiser to save energy, in both manufacture and sustainability. A better material in building wall applications can potentially “bridge the gap between the supply and demand of the energy”. In this review paper, the authors explained in detail their review of the phase change materials in building wall construction. Different types of phase change material were discussed. The challenges in the phase change material applications, such as phase segregation, stability, and thermal conductivity were reviewed. However, I suggest the authors consider addressing some minor concerns, listed below: 

  1. Figure 5. The scale bars in all images are not clear. And I suggest authors resize the images to make them the same size 
  2. Figure 10, please consider adding the scale bars for the 3 images on the right side 
  3. The authors discussed the inorganic phase change materials, I suggest authors add VO2 material in the inorganic section since the inorganic VO2 material is considered as a “smart” material for buildings. This material changes its phase upon external thermal excitation at around 67C which is useful in external walls. Below is a good reference that shows the technique to create this coating for large-scale, which could be a good addition to this review paper.  

https://doi.org/10.1016/j.tsf.2020.138117

  1. Figure 6, please add the figure legends to show the 0, 200, 400, 600, and 800 cycles 
  2. Figure 7b, the scale bar is hard to see, on a printed paper. Please consider updating the image

Author Response

This manuscript from Kurdi et al. shows their review of the phase change materials’ potential applications in the building wall construction. This is an important topic as the energy can be used much wiser to save energy, in both manufacture and sustainability. A better material in building wall applications can potentially “bridge the gap between the supply and demand of the energy”. In this review paper, the authors explained in detail their review of the phase change materials in building wall construction. Different types of phase change material were discussed. The challenges in the phase change material applications, such as phase segregation, stability, and thermal conductivity were reviewed. However, I suggest the authors consider addressing some minor concerns, listed below: 

  1. Figure 5. The scale bars in all images are not clear. And I suggest authors resize the images to make them the same size 

 Reply: Accepted. Visible scale bar was included in all the images.

  1. Figure 10, please consider adding the scale bars for the 3 images on the right side 

Reply: Accepted. Visible scale bar was included in all the images.

  1. The authors discussed the inorganic phase change materials, I suggest authors add VO2 material in the inorganic section since the inorganic VO2 material is considered as a “smart” material for buildings. This material changes its phase upon external thermal excitation at around 67C which is useful in external walls. Below is a good reference that shows the technique to create this coating for large-scale, which could be a good addition to this review paper. https://doi.org/10.1016/j.tsf.2020.138117

Reply: Accepted. Usage of VO2 as potential phase change material has been included in the revised manuscript with due ref. as suggested by the reviewer. Please see the highlighted text in the revised manuscript.

  1. Figure 6, please add the figure legends to show the 0, 200, 400, 600, and 800 cycles 

Reply: Accepted. Fig. legend has been included in Fig. 6.

  1. Figure 7b, the scale bar is hard to see, on a printed paper. Please consider updating the image

Reply: Accepted. Scale bar in Fig. 7 has been updated.

Reviewer 2 Report

The manuscript deals with a hot topic in Material Science: phase change materials in building wall construction. In fact, many papers and reviews related to this content are available in the literature (and suitably cited in the text by the authors) even though most are not recent.  So, I expect that the motivation of the authors should be to provide to the readers a survey of the most recent developments. By contrast, I found only 49 references cited among 190 (representing only slightly more than 25%) referred to papers published in the last 5 years (from 2016).   Organization of the content seems not clear to give a complete idea of the status of the art, but different paragraphs are shown without a unified scenario, like that found for histance in a recent review on the same topic: J. Lizana, R. Chacartegui, A. Barrios-Padura, C. Ortiz, Advanced low-carbon energy measures based on thermal energy storage in buildings: A review, Renewable and Sustainable Energy Reviews, 82​(3​)​ (2018​) 3705-3749,​ ​https://doi.org/10.1016/j.rser.2017.10.093.   Furthermore, the authors stated in the Introduction: "Nevertheless, it is difficult to have a good understanding because most of the reported work is not scientifically linked to previous work and is disorganized. There are also a number of reviews in the literature, but they are disjointed."​ So, a reader is expected to find a review paper in which all these questionable points have been overcome, but, in my opinion, this seems not the case.   In addition, all the figures seem to be taken (redrawn as stated often by the authors in their captions) from papers already published and cited properly, but I don't know if they asked permission for reproducing them to the corresponding Publishers.​   ​For all the above mentioned reason I can't recommend the publication of this manuscript in this journal and major revision is needed.​   ​The following minor changes deserve to be performed. In the whole text I found several times the number of chemical formulas often not reported as superscript (as it must be), like for CO2 in page 2 (instead of CO2). In other cases the number representing the exponent of units must be reported as a subscript (see MJ/m3 in page 8 instead of MJ/m3).   Title: Please, replace "-" with "."   Author's surnames: Please, delete "and *".   Fig. 1: Please, replace Standstone with sandstone.   End of page 9, second and third lines from the bottom: I found the sentence with an interruption. Please, check and revise.

Author Response

The manuscript deals with a hot topic in Material Science: phase change materials in building wall construction. In fact, many papers and reviews related to this content are available in the literature (and suitably cited in the text by the authors) even though most are not recent.  So, I expect that the motivation of the authors should be to provide to the readers a survey of the most recent developments. By contrast, I found only 49 references cited among 190 (representing only slightly more than 25%) referred to papers published in the last 5 years (from 2016).  

Reply: In view of reviewer’s suggestion a number of recent publications (2019-2020) on that was cited in the revised manuscript. The author hopes this will enrich the standard of the present manuscript. Please see the highlighted text in the revised manuscript.

Organization of the content seems not clear to give a complete idea of the status of the art, but different paragraphs are shown without a unified scenario, like that found for histance in a recent review on the same topic: J. Lizana, R. Chacartegui, A. Barrios-Padura, C. Ortiz, Advanced low-carbon energy measures based on thermal energy storage in buildings: A review, Renewable and Sustainable Energy Reviews, 82​(3​)​ (2018​) 3705-3749,​ ​https://doi.org/10.1016/j.rser.2017.10.093.  

Reply: The content and focus of the present manuscript is different than that of the one mentioned by the reviewer. The paper by Lizana et al focused on the broad aspect of the thermal energy storage in buildings, where as out current paper focused particularly in building wall application. Having said that, there could be some similarity in organization point of view, as PCM was the core subject in both cases.

Furthermore, the authors stated in the Introduction: "Nevertheless, it is difficult to have a good understanding because most of the reported work is not scientifically linked to previous work and is disorganized. There are also a number of reviews in the literature, but they are disjointed."​ So, a reader is expected to find a review paper in which all these questionable points have been overcome, but, in my opinion, this seems not the case.  

Reply: Accepted. The statement was an oversight and revised accordingly in the revised manuscript.

In addition, all the figures seem to be taken (redrawn as stated often by the authors in their captions) from papers already published and cited properly, but I don't know if they asked permission for reproducing them to the corresponding Publishers.​ ​

Reply: Yes, copyright permission was obtained as required and submitted to the editorial office during original submission of the manuscript.

For all the above mentioned reason I can't recommend the publication of this manuscript in this journal and major revision is needed.​ 

Reply: The manuscript was thoroughly revised in view of reviewers’ suggestion and improved accordingly. The authors hope the manuscript now hold the standard to be publishes in Materials (MDPI) journal. 

The following minor changes deserve to be performed. In the whole text I found several times the number of chemical formulas often not reported as superscript (as it must be), like for CO2 in page 2 (instead of CO2). In other cases, the number representing the exponent of units must be reported as a subscript (see MJ/m3 in page 8 instead of MJ/m3).   Title: Please, replace "-" with "."   Author's surnames: Please, delete "and *".   Fig. 1: Please, replace Standstone with sandstone.   End of page 9, second and third lines from the bottom: I found the sentence with an interruption. Please, check and revise.

Reply: Accepted. All those typos were corrected in the revised manuscript. Please see the highlighted text in the revised manuscript.

Reviewer 3 Report

The paper deals with an interesting topics and is well presented. 

I have the following comments:

1-figure 1; can authors give more details? what does conventional thermal mass mean?

2-section 3.3.3 do authors mean hydrated instead of hydrogenated?

3-section 3.3.3:   sentence is  incomplete:

Physical state of some common hydrated salts...

4-  section 4.1 authors said that the extra water pricniple was described in detail earlier....it was briefly cited in 3.3.3. and not detailed!!!

section 4.1  title: phase segregation and subcooling

authors speak about supercooling....

can you add definitions to subcooling and supercooling

5- figure 17 can you give some benefits from adding PCM to bricks; any comparisons, numbers?

6- what does the unit EJ stands for?

Author Response

The paper deals with an interesting topics and is well presented. 

I have the following comments:

1-figure 1; can authors give more details? what does conventional thermal mass mean?

Reply: Accepted. Additional text has been added on that. Please see the highlighted text in the revised manuscript.

2-section 3.3.3 do authors mean hydrated instead of hydrogenated?

Reply: Accepted. It was an oversight and corrected accordingly in the revised manuscript. Please see the highlighted text in the revised manuscript.

3-section 3.3.3:   sentence is  incomplete:

Physical state of some common hydrated salts...

Reply: Accepted. It was an oversight and deleted in the revised manuscript.

4-  section 4.1 authors said that the extra water pricniple was described in detail earlier....it was briefly cited in 3.3.3. and not detailed!!!

Reply: Accepted. It was an oversight and corrected accordingly in the revised manuscript. Please see the highlighted text in the revised manuscript.

section 4.1  title: phase segregation and subcooling

authors speak about supercooling....

can you add definitions to subcooling and supercooling

Reply: Accepted. The section title was revised and additional text regarding ‘supercooling’ was included in the revised manuscript. Please see the highlighted text in the revised manuscript.

5- figure 17 can you give some benefits from adding PCM to bricks; any comparisons, numbers?

Reply: Accepted. Additional information on that, with numerical values to compare the effect of PCM addition on hollow bricks has been included in the revised manuscript. Please see the highlighted text in the revised manuscript.

6- what does the unit EJ stands for?

Reply: EJ stands for ‘exajoule’ which is an SI unit of for energy and equal to 1018 joules. It was clarified in the revised manuscript. Please see the highlighted text in the revised manuscript.

Reviewer 4 Report

The study entitled Potential phase change materials in building wall construction-A review, presents a critical evaluation of PCMs , focusing on two aspects: (i) PCMs for building wall applications and (ii) inclusion of PCMs in building wall applications. My general comments are as follows:

  1. The methodology section is missing. The authors should include a section in which they describe their methodology.
  2. The organization of the paper suffers. The rational of the sequence of the consecutive sections is not clear, neither their relation. The authors should restructure their material.
  3. A significant aspect which is not considered in this study is related to the orientation of the integration of PCM into building elements - the authors are suggested to advise  Construction and Building Materials225, 452-464.
  4. Would you explicitly specify the novelty of your work? What progress against the most recent state-of-the-art similar studies was made?
  5. Please improve the graphics quality (eg figures 1, 6 and 19)
  6. The conclusions part should be enriched with further findings of the study.
  7. Large tables are hard to read. Please consider including the content of eg Table 6 in more than one tables.
  8. There are several sentence structure problems and grammatical errors and hence hampering understandability and comprehension of the manuscript. Besides this, a long string of citations does not really add value to the manuscript.

Overall, the reviewer humbly feels that the manuscript still has much room for improvement, even though it's reasonably well-written. 

Author Response

The study entitled Potential phase change materials in building wall construction-A review, presents a critical evaluation of PCMs , focusing on two aspects: (i) PCMs for building wall applications and (ii) inclusion of PCMs in building wall applications. My general comments are as follows:

  1. The methodology section is missing. The authors should include a section in which they describe their methodology.

Reply: As this was a review paper, thus ‘methodology section’ was not included. Some papers, as evident in literature, do include ‘methodology section’ if the focus of the work given on statistical analysis purpose, such as how many times a particular keyword was cited in the given publication period etc. Since this (statistical analysis) was not the focus of the present manuscript, thus methodology section was avoided. Having said that, details fabrication of PCMs was included in respective sections.

  1. The organization of the paper suffers. The rational of the sequence of the consecutive sections is not clear, neither their relation. The authors should restructure their material.

Reply: In view of reviewer suggestion, the whole manuscript was revised thoroughly. Efforts were made for better connectivity and consecutively among different sections. In view of that, the authors hope the manuscript now hold the ground to be considered to publish in Materials journal.

  1. A significant aspect which is not considered in this study is related to the orientation of the integration of PCM into building elements - the authors are suggested to advise  Construction and Building Materials225, 452-464.

Reply: Accepted. Additional text on that was included with the suggested reference from the reviewer. Please see the highlighted text in the revised manuscript.

  1. Would you explicitly specify the novelty of your work? What progress against the most recent state-of-the-art similar studies was made?

Reply: Additional text regarding the novelty of the present work was explicitly specified in the introduction section. Against the most recent state-of-the-art similar studies. The present study mainly focuses on the inclusion of PCMs in building wall application. As stated in the introduction section of the revised manuscript, due to the construction of mage projects and building in the Middle East region, particularly in Saudi Arabia, there is a huge demand for PCMs in building wall applications. The current studies will enable engineers to select the particular PCMs for a given application as well as help the researchers to carry out more innovative research work in this area. Please see the highlighted text in the revised manuscript.

  1. Please improve the graphics quality (eg figures 1, 6 and 19)

Reply: Accepted and Figures 1 and 6 were updated accordingly.

  1. The conclusions part should be enriched with further findings of the study.

Reply: Additional text was included in conclusion section in view of the objective of the present work to enrich it. Please see the highlighted text in the revised manuscript.

  1. Large tables are hard to read. Please consider including the content of eg Table 6 in more than one tables.

Reply: Accepted Table 6 has been restructured for better readability. Please see the highlighted text in the revised manuscript.

There are several sentence structure problems and grammatical errors and hence hampering understandability and comprehension of the manuscript. Besides this, a long string of citations does not really add value to the manuscript.

Reply: The manuscript was revised carefully by native English speaker and effort was taken to avoid typos, spelling and grammatical errors as much as possible.

As this was a review paper and lots of information were required to be condensed in scientific form, thus it was required to include lots of references. This will not only make the present work creditable but also give the opportunity to the read to dig deep in their respective area of interest.

Overall, the reviewer humbly feels that the manuscript still has much room for improvement, even though it's reasonably well-written. 

Reply: Thanks for reviewer’s positive and constructive suggestions.

Round 2

Reviewer 2 Report

The authors revised thei manuscript and I may recognize that a slight improvement is clearly evident, but many points were not addressed. In particular, the following aspects need to be clarified.   To be honest, the addition of only 6 papers cited does not seem to significantly change the situation. Therefore, I cannot conclude that the R1 version may provide to the readers a survey of the most recent developments, as most of the readers should expect. In the following revision a more significant addition of recent papers published in the last five years (no less than 10 papers more) must be added.   The Organization of the content was not modified (as I suggested previously), even though I can recognize that the current manuscript is different from the one I mentioned in the last review. A didn't see improvement in clarity in the current version. They must reorganize the content to achieve this goal.   Please, check and revise the symbol representing the temperature unit: degree is represented by ° without any line (see end of page 12 of 43).   End of page 22 of 43: Replace "In other word" with "In other words". In addition, please, put "2" of "m2" in superscript.   In conclusion, I ask the authors to make further efforts to address the remaining questionable points and minor revision is recommended to this end.

Author Response

The authors revised their manuscript and I may recognize that a slight improvement is clearly evident, but many points were not addressed. In particular, the following aspects need to be clarified. To be honest, the addition of only 6 papers cited does not seem to significantly change the situation. Therefore, I cannot conclude that the R1 version may provide to the readers a survey of the most recent developments, as most of the readers should expect. In the following revision a more significant addition of recent papers published in the last five years (no less than 10 papers more) must be added.

Reply: Accepted. As per reviewers suggestion, 13 additional papers published in the last five years were referred in the revised version of the manuscript. Please see the green highlighted text in the revised manuscript.

The Organization of the content was not modified (as I suggested previously), even though I can recognize that the current manuscript is different from the one I mentioned in the last review. A didn't see improvement in clarity in the current version. They must reorganize the content to achieve this goal.

Reply: At this point, any further organization of the paper is out of scope, as rest (majority) of the reviewers are content with current organization of the paper, as per their respective review reports.

Please, check and revise the symbol representing the temperature unit: degree is represented by ° without any line (see end of page 12 of 43). End of page 22 of 43: Replace "In other word" with "In other words". In addition, please, put "2" of "m2" in superscript.

Reply: Accepted and revised accordingly. Please see the highlighted text in the revised manuscript.

In conclusion, I ask the authors to make further efforts to address the remaining questionable points and minor revision is recommended to this end.

Reply: The manuscript was thoroughly revised in view of ALL of the reviewers’ suggestion and improved accordingly. The authors hope the manuscript now hold the standard to be publishes in Materials (MDPI) journal. 

Reviewer 4 Report

All my comments were adequately addressed.

Author Response

Authors sincere thanks for reviewer’s positive feedback.